# TRACE: Theoretical Risk Attribution under Covariate-shift Effects

## Abstract

When a source-trained model $Q$ is replaced by a model $\tilde{Q}$ trained on shifted data, its performance on the source domain can change unpredictably. To address this, we study the two-model risk change, $\Delta R := R_P(Q) - R_P(\tilde{Q})$, under covariate shift. We introduce TRACE (Theoretical Risk Attribution under Covariate-shift Effects), a framework that decomposes $|\Delta R|$ into an interpretable upper bound. This decomposition disentangles the risk change into four actionable factors: two generalization gaps, a model change penalty, and a covariate shift penalty, transforming the bound into a powerful diagnostic tool for understanding why performance has changed. To make TRACE a fully computable diagnostic, we instantiate each term. The covariate shift penalty is estimated via a model sensitivity factor (from high-quantile input gradients) and a data-shift measure; we use feature-space Optimal Transport (OT) by default and provide a robust alternative using Maximum Mean Discrepancy (MMD). The model change penalty is controlled by the average output distance between the two models on the target sample. Generalization gaps are estimated on held-out data. We validate our framework in an idealized linear regression setting, showing the TRACE bound correctly captures the scaling of the true risk difference with the magnitude of the shift. Across synthetic and vision benchmarks, TRACE diagnostics are valid and maintain a strong monotonic relationship with the true performance degradation. Crucially, we derive a deployment gate score from the model change and covariate shift terms that strongly correlates with $|\Delta R|$ and achieves exceptionally high AUROC/AUPRC for gating decisions, enabling safe, label-efficient model replacement.

## 1 Introduction

Replacing a production model $Q$ with an updated model $\tilde{Q}$ trained on shifted data is a routine yet high-stakes operation. Even under pure covariate shift, where the labeling function is invariant ($P(y \mid x) = \tilde{P}(y \mid \tilde{x})$), the new model's performance on the original source domain can change unpredictably. This challenge is not merely academic; such shifts are ubiquitous in modern deployment, arising from sensor upgrades, regional expansion, or temporal drift. Consider a concrete example: a medical imaging model originally trained at Hospital A is updated using data from Hospital B's new scanners. For safety, the new model must not degrade performance on Hospital A's legacy patient population. In this and many other cases, the source domain acts as a mission-critical "anchor." A new model that improves on the shifted data but silently harms this anchor distribution represents a significant safety failure. While measuring the net performance change is straightforward, explaining *why* it changed is not. Practitioners need a diagnostic tool that can attribute the change to its root causes: is it due to the geometric data shift, the model instability during retraining, or simple generalization error?

To this end, we introduce TRACE (Theoretical Risk Attribution under Covariate-shift Effects), a framework for analyzing the two-model risk change, $\Delta R := R_P(Q) - R_P(\tilde{Q})$, on the fixed source distribution $P$. At the heart of TRACE is a conceptual decomposition that provides an attribution lens for this risk change. Under a mild Lipschitz condition, we can bound the absolute change $|\Delta R|$:

$$|\Delta R| \leq \underbrace{G_Q}_{\substack{\text{Source} \\ \text{Gen. Gap}}} + \underbrace{G_{\tilde{Q}}}_{\substack{\text{Target} \\ \text{Gen. Gap}}} + \underbrace{D_{Q,\tilde{Q}}}_{\substack{\text{Empirical} \\ \text{Discrepancy}}} + \underbrace{\text{COSP}}_{\substack{\text{Covariate} \\ \text{Shift Penalty}}} \qquad (1)$$

This inequality disentangles the risk change into four interpretable factors. The algebraic steps are simple (triangle inequality and Kantorovich-Rubinstein duality); the novelty lies in using this decomposition as a two-model attribution framework on a source domain. This perspective separates the geometric effects of data shift from the algorithmic effects of retraining, a necessary lens for safe model replacement that complements classic single-model transfer bounds.

Our main contribution is to operationalize this conceptual lens into a fully computable, high-probability diagnostic report. We estimate the geometric data shift using feature-space Optimal Transport (OT) or Maximum Mean Discrepancy (MMD). We further split the empirical discrepancy, controlling the model change component via an average output distance on the target sample. Model sensitivity is captured by robust Lipschitz proxies derived from high-quantile input-gradient norms. Combining these estimators with standard concentration inequalities yields a non-asymptotic, end-to-end diagnostic with a full attribution breakdown.

Crucially, TRACE is designed as a practical *tool* for practitioners. Each term in the decomposition is paired with an actionable interpretation: large generalization gaps suggest overfitting and motivate regularization; a large model-change term points to unstable fine-tuning or aggressive hyperparameters; and a large covariate-shift penalty signals that the input distribution has moved far from the anchor.

The practical value of TRACE is demonstrated through a simplified deployment gate score, derived from its core components: model change and covariate shift. Across synthetic and vision benchmarks, this score strongly correlates with the true performance degradation. We also show that, in vision benchmarks, it reliably ranks potentially harmful updates and achieves exceptionally high AUROC/AUPRC for gating decisions. TRACE thus provides an actionable framework to understand, predict, and control the risks of model replacement, enabling safer and more label-efficient machine learning deployment.

## 2 RELATED WORK

Our work introduces an attribution framework for the two-model risk change that occurs during model replacement under covariate shift. This goal is distinct from, but related to, several established lines of research, most notably the theory of single-model domain adaptation. We situate TRACE by highlighting these connections and differences.

**Domain Adaptation and Divergence Measures.** Domain adaptation (DA) theory primarily aims to bound the target-domain risk of a *single model* trained on a source domain (Ben-David et al., 2010; Mansour et al., 2009). Such bounds rely on divergence terms to quantify distribution shift, with recent work leveraging geometry-aware metrics like the Wasserstein distance from Optimal Transport (OT)(Redko et al., 2017; Courty et al., 2017; Shen et al., 2018) and feature-space distances like Maximum Mean Discrepancy (MMD)(Gretton et al., 2012; Long et al., 2015; Tzeng et al., 2014). Other discrepancy-based approaches have explored margin disparities (Zhang et al., 2019) and $Y$-discrepancy (Cortes et al., 2010; Mohri & Medina, 2012), establishing a rich landscape of divergence measures. TRACE fundamentally differs from this classic setup in its goal. Instead of bounding the target risk $R_{\tilde{P}}(Q)$, TRACE analyzes the *risk change on the source domain* between two models, $|R_P(Q) - R_P(\tilde{Q})|$. This addresses the practical MLOps problem of *safe model replacement*, where the priority is to ensure an update does not degrade performance on a critical anchor distribution. While we leverage standard tools like OT and MMD, we repurpose them for attribution within this novel two-model, source-centric decomposition.

**Model Disagreement and Algorithmic Stability.** Differences between models are an informative signal in several fields. Disagreement-based DA methods, for example, use prediction differences for unsupervised adaptation (Saito et al., 2017; 2018). Separately, algorithmic stability theory analyzes how training perturbations affect the learned model to provide generalization guarantees (Bousquet & Elisseeff, 2002; Hardt et al., 2016; Kuzborskij & Lampert, 2018). TRACE repurposes a related idea—the average output distance between models—not to drive adaptation or bound generalization, but to quantify the *model change* component within our risk attribution framework.

**Generalization Under Distribution Mismatch.** Other theoretical works bound the single-learner generalization error under distribution mismatch using tools from PAC-Bayesian theory (Germain et al., 2016) and information theory (Russo & Zou, 2016; Xu & Raginsky, 2017; Steinke & Zakynthinou, 2020). While these works aim to bound the generalization gap itself, TRACE is complementary: it situates the generalization gaps alongside other quantities in a unified inequality that explains the total change in performance.

**Shift Detection and Monitoring.** The problem of dataset shift is well-documented, with taxonomies distinguishing covariate, label, and conditional shift (Quiñonero-Candela et al., 2009; Lipton et al., 2018). In practice, shift is often addressed by monitoring methods that detect its occurrence via two-sample tests (Rabanser et al., 2019) or out-of-distribution detectors (Hendrycks & Gimpel, 2016). These methods typically produce an alarm *rather than an explanation*. TRACE provides the next step: a quantitative attribution connecting a detected shift to its impact on model risk.

## 3 PROBLEM SETUP

**Setting and Goal.** We address the problem of replacing a model in the presence of data shift. We consider a source distribution $P$ and a target distribution $\tilde{P}$ over a common space $\mathcal{X} \times \mathcal{Y}$. We operate under the standard **covariate shift** assumption, where the input distributions differ but the conditional label distribution remains invariant:

$$P(y \mid x) = \tilde{P}(y \mid \tilde{x}), \qquad P_X \neq \tilde{P}_X.$$

A source model $Q$, with predictor $f_Q : \mathcal{X} \to \mathbb{R}^C$, is trained on an i.i.d. sample $S = \{(x_i, y_i)\}_{i=1}^n$ from $P$. A target model $\tilde{Q}$, with predictor $f_{\tilde{Q}}$, is trained on an i.i.d. sample $\tilde{S} = \{(\tilde{x}_i, \tilde{y}_i)\}_{i=1}^n$ from $\tilde{P}$. Our goal is to analyze the change in risk on the original **source domain** $P$ when replacing $Q$ with $\tilde{Q}$. We formally study the two-model risk change[1],

$$\Delta R := R_P(Q) - R_P(\tilde{Q}),$$

and seek to attribute its magnitude $|\Delta R|$ to distinct factors such as generalization error, data shift, and model change.

**Loss and Risks.** Let $\ell : \mathbb{R}^C \times \mathcal{Y} \to \mathbb{R}^+$ be a loss function. We define the population risk of a model $Q$ on a distribution $P$ and its corresponding empirical risk on a sample $S$ as:

$$R_P(Q) = \mathbb{E}_{(x,y) \sim P}[\ell(f_Q(x), y)], \quad \widehat{R}_S(Q) = \frac{1}{n} \sum_{i=1}^n \ell(f_Q(x_i), y_i).$$

The risks $R_{\tilde{P}}(\tilde{Q})$ and $\widehat{R}_{\tilde{S}}(\tilde{Q})$ are defined analogously.

**Quantities for Attribution.** Our analysis decomposes $|\Delta R|$ using the following key quantities: the source generalization gap $G_Q$, the target generalization gap $G_{\tilde{Q}}$, and the empirical discrepancy $D_{Q,\tilde{Q}}$. These are defined as:

$$G_Q := |R_P(Q) - \widehat{R}_S(Q)|, \quad G_{\tilde{Q}} := |R_{\tilde{P}}(\tilde{Q}) - \widehat{R}_{\tilde{S}}(\tilde{Q})|, \quad D_{Q,\tilde{Q}} := |\widehat{R}_S(Q) - \widehat{R}_{\tilde{S}}(\tilde{Q})|.$$

As we show in Section 4, the empirical discrepancy $D_{Q,\tilde{Q}}$ is further split to isolate the effects of data shift from those of the model update.

**Assumptions.** We assume the input space $(\mathcal{X}, d_{\mathcal{X}})$ is a Polish metric space and that the marginals $P_X, \tilde{P}_X$ have finite first moments, ensuring the Wasserstein distance is well defined. Our framework relies on the following standard assumptions:

**Assumption 1** (Bounded Loss). *The loss function $\ell(z, y)$ is bounded in $[0, M]$. For multiclass cross-entropy with $C$ classes and logits $z = f(x)$, enforcing a bound on the logits such as $\|z\|_\infty \leq B$ (e.g., via clipping) guarantees this, with $M = B + \log C$. This is a standard practice in production ML for numerical stability and has negligible impact on model accuracy, while being essential for the concentration bounds we use.*

---

[1]We chose the convention $\Delta R = R_P(Q) - R_P(\tilde{Q})$ so that a positive value represents an **improvement** (reduction) in risk.

**Assumption 2** (Input Lipschitz Continuity of the Loss). *For any predictor $f$, the end-to-end loss function is $L_x(f)$-Lipschitz with respect to the input $x$. That is, there exists a constant $L_x(f) > 0$ such that for all $x, x' \in \mathcal{X}$ and any $y \in \mathcal{Y}$:*

$$|\ell(f(x), y) - \ell(f(x'), y)| \le L_x(f)\, d_\mathcal{X}(x, x') \quad \text{for all } x, x', y.$$

*In practice, this smoothness property is naturally encouraged by common regularization techniques such as weight decay.*

**Assumption 3** (Logit Lipschitz Continuity of the Loss). *The loss function is $L_\ell$-Lipschitz with respect to its logit argument $z$. That is, there exists a constant $L_\ell > 0$ such that for all $z, z' \in \mathbb{R}^C$, any $y \in \mathcal{Y}$, and a norm $\|\cdot\|$ on $\mathbb{R}^C$:*

$$|\ell(z, y) - \ell(z', y)| \le L_\ell \|z - z'\| \quad \text{for all } z, z', y.$$

*This is a standard and mild assumption. For the widely-used softmax cross-entropy loss, this condition holds globally with a constant $L_\ell \le \sqrt{2}$ for the $\ell_2$ norm, requiring no special architectural changes for typical deep learning models.*

**General Applicability.** The TRACE framework is not restricted to classification. The entire derivation holds for any bounded, Lipschitz loss, including regression problems (typically using a bounded variant like the Huber loss). We demonstrate this explicitly in Section 5 with a detailed analysis of Ridge Regression.

**Wasserstein Distance.** We primarily measure covariate shift using the 1-Wasserstein distance.

**Definition 1** (1-Wasserstein). *The 1-Wasserstein distance between distributions $\mu$ and $\nu$ is*

$$\mathrm{W}_1(\mu, \nu) := \inf_{\pi \in \Pi(\mu, \nu)} \int d_\mathcal{X}(x, x')\, d\pi = \sup_{\|g\|_{\mathsf{Lip}} \le 1} \{\mathbb{E}_\mu[g] - \mathbb{E}_\nu[g]\}.$$

The dual formulation, based on Kantorovich-Rubinstein duality, is central to our analysis as it connects the distance between distributions to the behavior of Lipschitz functions (Villani, 2009, Thm. 6.15). We leverage this property to bound the impact of data shift on a model's risk.

## 4 FROM DECOMPOSITION TO A COMPUTABLE TRACE DIAGNOSTIC

Our goal is to transform the abstract risk change $|\Delta R|$ into a practical diagnostic. We begin by decomposing it into four high-level components. This initial decomposition serves as the scaffold for our analysis, with subsequent subsections detailing how each theoretical term is controlled and instantiated using specific divergence measures.

### 4.1 A GENERAL FOUR-TERM DECOMPOSITION

The foundation of TRACE is a general inequality that connects the true risk change to a set of more tractable quantities.

**Lemma 1** (General TRACE Decomposition). *The absolute risk change is bounded as follows:*

$$|\Delta R| = \left| R_P(Q) - R_P(\tilde{Q}) \right| \le G_Q + D_{Q, \tilde{Q}} + G_{\tilde{Q}} + \mathsf{COSP}.$$

*where* $\mathsf{COSP} := |R_{\tilde{P}}(\tilde{Q}) - R_P(\tilde{Q})|$ *represents the population-level risk change for a fixed model $\tilde{Q}$ due to the shift from $P_X$ to $\tilde{P}_X$.*

The proof of this lemma, detailed in Appendix B, follows from a direct application of the triangle inequality by inserting the empirical risks $\widehat{R}_S(Q)$ and $\widehat{R}_{\tilde{S}}(\tilde{Q})$. The core of our framework lies in how we bound and estimate the $\mathsf{COSP}$ and $D_{Q, \tilde{Q}}$ terms. The $\mathsf{COSP}$ term can be controlled using various divergence measures. Our primary approach, detailed throughout this section, is based on Optimal Transport (OT). Under Assumption 2, Kantorovich-Rubinstein duality yields the well-known OT-based instantiation:

$$\mathsf{COSP} = |R_{\tilde{P}}(\tilde{Q}) - R_P(\tilde{Q})| \le L_x(f_{\tilde{Q}}) W_1(P_X, \tilde{P}_X). \tag{2}$$

We also present a robust alternative based on Maximum Mean Discrepancy (MMD) in Section 4.6. The remainder of this section focuses on making the OT-based diagnostic fully computable.

## 4.2 CONTROLLING THE EMPIRICAL DISCREPANCY

The empirical discrepancy term, $D_{Q,\tilde{Q}} = |\widehat{R}_S(Q) - \widehat{R}_{\tilde{S}}(\tilde{Q})|$, encapsulates two distinct effects: the difference between the source and target *samples*, and the difference between the source and target *models*. To isolate these, we split $D_{Q,\tilde{Q}}$ using the triangle inequality:

$$D_{Q,\tilde{Q}} \leq \underbrace{|\widehat{R}_S(Q) - \widehat{R}_{\tilde{S}}(Q)|}_{\text{Empirical Data Shift}} + \underbrace{|\widehat{R}_{\tilde{S}}(Q) - \widehat{R}_{\tilde{S}}(\tilde{Q})|}_{\text{Model Change}}. \tag{3}$$

This split enables label-efficient, high-probability control of each component. The first term, which measures the effect of data shift on a fixed model, can be controlled via transport distances. The second term, which measures the effect of the model update, can be controlled via the distance between model outputs. This separation provides clear guidance on whether to intervene on the data or on the retraining process itself. We now provide formal bounds for each term.

The model change term is controlled by the average distance between the models' outputs, a measure of *algorithmic instability*. The proof follows from a direct application of the triangle inequality and Assumption 3.

**Proposition 1** (Model Change Bound). *Under Assumption 3 (Logit Lipschitz Continuity), the model change term is bounded by:*

$$\left|\widehat{R}_{\tilde{S}}(Q) - \widehat{R}_{\tilde{S}}(\tilde{Q})\right| \leq \frac{L_\ell}{n} \sum_{i=1}^{n} \left\| f_Q(\tilde{x}_i) - f_{\tilde{Q}}(\tilde{x}_i) \right\|. \tag{4}$$

The term for empirical data shift is controlled by the geometric distance between the samples, plus a statistical remainder for label noise.

**Lemma 2** (Empirical Shift Bound). *Under Assumptions 1 and 2, for any $\delta \in (0,1)$, the following holds with probability at least $1 - \delta$ over draws of $S$ and $\tilde{S}$:*

$$\left|\widehat{R}_S(Q) - \widehat{R}_{\tilde{S}}(Q)\right| \leq L_x(f_Q) W_1(\widehat{P}_n, \widehat{\widetilde{P}}_n) + 2M\sqrt{\frac{1}{2n}\log\frac{4}{\delta}}, \tag{5}$$

*where $\widehat{P}_n$ and $\widehat{\widetilde{P}}_n$ are the empirical distributions of the inputs in $S$ and $\tilde{S}$, respectively.*

The proof (Appendix B) uses Kantorovich-Rubinstein duality to control the effect of the input shift and Hoeffding's inequality to bound the finite-sample noise from the labels.

## 4.3 FROM POPULATION QUANTITIES TO COMPUTABLE PROXIES

The bounds derived so far involve unobservable population quantities, such as the true Wasserstein distance $W_1(P_X, \tilde{P}_X)$ and the global Lipschitz constants $L_x(f)$. To build a practical diagnostic, we must replace these with reliable, computable estimators based on the available data.

**Controlling the Population Wasserstein Distance (high probability).** The true distance $W_1(P_X, \tilde{P}_X)$ is unknown, but we can relate it to the empirical $W_1(\widehat{P}_n, \widehat{\widetilde{P}}_{\tilde{n}})$ via a triangle inequality together with finite-sample concentration of empirical measures. Fix $\delta \in (0,1)$ and let $\alpha_d := \max\{d, 2\}$. Under finite first–moment and mild tail/moment conditions in $\mathbb{R}^d$ (as in (Fournier & Guillin, 2015; Weed & Bach, 2019)), there exist distribution-dependent constants $C_X, C_{\tilde{X}} > 0$ such that, with probability at least $1 - \delta$,

$$W_1(P_X, \tilde{P}_X) \leq W_1(\widehat{P}_n, \widehat{\widetilde{P}}_{\tilde{n}}) + \underbrace{C_X \left(\frac{\log(4/\delta)}{n}\right)^{1/\alpha_d}}_{:= \varepsilon_n(\delta)} + \underbrace{C_{\tilde{X}} \left(\frac{\log(4/\delta)}{\tilde{n}}\right)^{1/\alpha_d}}_{:= \tilde{\varepsilon}_{\tilde{n}}(\delta)}. \tag{6}$$

Fix $\delta \in (0,1)$ and let $\alpha_d := \max\{d, 2\}$ (the dimension–dependent exponent appearing in standard $W_1$ concentration inequalities). Equivalently, one can use the tail form $\Pr\left(W_1(P_X, \widehat{P}_n) > \epsilon\right) \leq A_X \exp\{-B_X n \epsilon^{\alpha_d}\}$ (and analogously for $\tilde{P}_X$), which yields equation 6 by choosing $\varepsilon_n(\delta), \tilde{\varepsilon}_{\tilde{n}}(\delta)$ accordingly and applying a union bound. In practice we compute $W_1(\widehat{P}_n, \widehat{\widetilde{P}}_{\tilde{n}})$ (in $x$ or in a feature space $h(x)$) as the main term and treat $\varepsilon_n, \tilde{\varepsilon}_{\tilde{n}}$ as explicit, vanishing remainder terms.

**Feature-Space Transport for Stability and Semantics.** Computing transport directly on high-dimensional raw inputs (e.g., pixels) is often computationally unstable and semantically meaningless. We therefore compute distances in a more meaningful feature space defined by a map $h : \mathcal{X} \to \mathbb{R}^k$. Assuming the map $h$ is bi-Lipschitz, we can upper- and lower-bound the Wasserstein distance with its feature-space counterpart, and therefore use it as a stable surrogate. In our practical diagnostic, we use the feature-space distance $W_1^{(h)}(\widehat{P}_n, \widehat{\tilde{P}}_n) = W_1(h\#\widehat{P}_n, h\#\widehat{\tilde{P}}_n)$ and introduce a representation scale factor $c_h$ to account for the mapping. We treat $c_h$ as a reported hyperparameter (defaulting to 1 for normalized features). For implementation, we use the fast and stable debiased Sinkhorn divergence as our proxy for the Wasserstein distance (Cuturi, 2013; Feydy et al., 2019). For implementation, we use the fast and stable debiased Sinkhorn divergence as our proxy for the Wasserstein distance (see Appendix F.2 for computational details and scalability).

**Estimating Lipschitz Constants with Gradient Norm Proxies.** The global input-Lipschitz constant $L_x(f) = \sup_{x,y} \|\nabla_x \ell(f(x), y)\|$ is computationally intractable. We instead estimate it using a robust, data-driven proxy. On a held-out set of $m$ examples, we compute the empirical distribution of input-gradient norms and take the $q$-quantile as our estimate:

$$\widehat{L}_x^{(q)}(f) := \mathrm{Quantile}_q \left( \{\|\nabla_x \ell(f(x_i), y_i)\|\}_{i=1}^m \right).$$

This high-quantile proxy (e.g., $q = 0.99$) provides a practical worst-case estimate of the model's sensitivity on real data, while being robust to outliers. The Dvoretzky-Kiefer-Wolfowitz (DKW) inequality (Massart, 1990) provides a finite-sample guarantee on the quality of this estimate. With probability at least $1 - \eta$, our empirical quantile $\widehat{L}_x^{(q)}(f)$ lies within an $\epsilon$-band of the true quantile $L_x^{(q)}(f)$, where $\epsilon = \sqrt{\frac{1}{2m} \log \frac{2}{\eta}}$.

### 4.4 THE PRACTICAL TRACE DIAGNOSTIC

We now assemble the components from the preceding subsections to present the final, fully computable TRACE diagnostic. This is a high-probability bound on the total risk change $|\Delta R|$, where each theoretical quantity has been replaced by its practical, data-driven proxy.

**Corollary 1** (Computable TRACE Diagnostic). *Fix confidence levels $\delta, \eta \in (0, 1)$. With probability at least $1 - \delta - \eta$ over the random draws of the training sets $S, \tilde{S}$ and any held-out validation sets, the following bound holds, conditioned on the event that the gradient-quantile proxies are valid upper bounds for the true Lipschitz constants (i.e., $\widehat{L}_x^{(q)}(f) \geq L_x(f)$ for $f \in \{f_Q, f_{\tilde{Q}}\}$):*

$$|\Delta R| \leq \underbrace{\widehat{G}_Q^{\mathrm{val}} + \widehat{G}_{\tilde{Q}}^{\mathrm{val}}}_{\text{Validation Gaps}} + \underbrace{\frac{L_\ell}{n} \sum_{i=1}^n \|f_Q(\tilde{x}_i) - f_{\tilde{Q}}(\tilde{x}_i)\|}_{\text{Model Change (Output Distance)}}$$

$$+ \underbrace{\left( \widehat{L}_x^{(q)}(f_Q) + \widehat{L}_x^{(q)}(f_{\tilde{Q}}) \right) c_h W_1^{(h)}(\widehat{P}_n, \widehat{\tilde{P}}_n)}_{\text{Empirical Shift Penalty}} \qquad (7)$$

$$+ \underbrace{2M \sqrt{\frac{1}{2n} \log \frac{4}{\delta}}}_{\text{Label Noise Remainder}} + \underbrace{M \left( \sqrt{\frac{1}{2m} \log \frac{2}{\eta}} + \sqrt{\frac{1}{2\tilde{m}} \log \frac{2}{\eta}} \right)}_{\text{Validation Set Error}} + \underbrace{\widehat{L}_x^{(q)}(f_{\tilde{Q}}) c_h \varepsilon_n}_{\text{Population Residual}}.$$

*All other quantities are defined in the preceding subsections. The validation gap estimators are defined as $\widehat{G}_Q^{\mathrm{val}} := |\widehat{R}_V(Q) - \widehat{R}_S(Q)|$ and $\widehat{G}_{\tilde{Q}}^{\mathrm{val}} := |\widehat{R}_{\tilde{V}}(\tilde{Q}) - \widehat{R}_{\tilde{S}}(\tilde{Q})|$, where $V \subset S$ and $\tilde{V} \subset \tilde{S}$ are validation splits of size $m$ and $\tilde{m}$, respectively.*

*Proof.* The proof is a direct assembly. We start with the main decomposition from Lemma 1. We bound the empirical discrepancy term $D_{Q,\tilde{Q}}$ using the split in equation 3, followed by Proposition 1 and Lemma 2. We control the population shift penalty using the bound in equation 6. Finally, we replace all theoretical quantities ($L_x$, $W_1$, etc.) with their computable proxies defined in Section 4.3. The high-probability statement follows from a union bound over the individual probabilistic guarantees. $\square$

**Interpretation.** This theorem provides a calibrated, conservative diagnostic. The final bound, which we denote $\widehat{\mathcal{B}}$, is controlled with high probability. While the Lipschitz factors rely on high-quantile proxies rather than certified global bounds, the overall diagnostic provides a reliable estimate of the scale of the risk change and, more importantly, an interpretable attribution of its sources.

**On the two Lipschitz factors.** A key feature of the bound is the term $(\widehat{L}_x^{(q)}(f_Q) + \widehat{L}_x^{(q)}(f_{\tilde{Q}}))$ that multiplies the empirical shift distance $W_1^{(h)}$. This arises because the total risk change involves two distinct "shift" effects that are combined through the triangle inequality. The constant $L_x(f_Q)$ is needed to control the *empirical shift*, $|\widehat{R}_S(Q) - \widehat{R}_{\tilde{S}}(Q)|$, which depends on the sensitivity of the source model. The constant $L_x(f_{\tilde{Q}})$ is needed to control the *population shift*, $|R_{\tilde{P}}(\tilde{Q}) - R_P(\tilde{Q})|$, which depends on the sensitivity of the target model. Both contribute to the final bound on the discrepancy measured by $W_1^{(h)}$.

### 4.5 THE TRACE DIAGNOSTIC PIPELINE IN PRACTICE

We now summarize the entire process of computing the practical TRACE diagnostic, $\widehat{\mathcal{B}}$, from Corollary 1. The pipeline, illustrated in Figure 1, is a sequence of parallel estimation steps that feed into a final aggregation stage. This flowchart provides a self-contained summary of the full diagnostic computation.

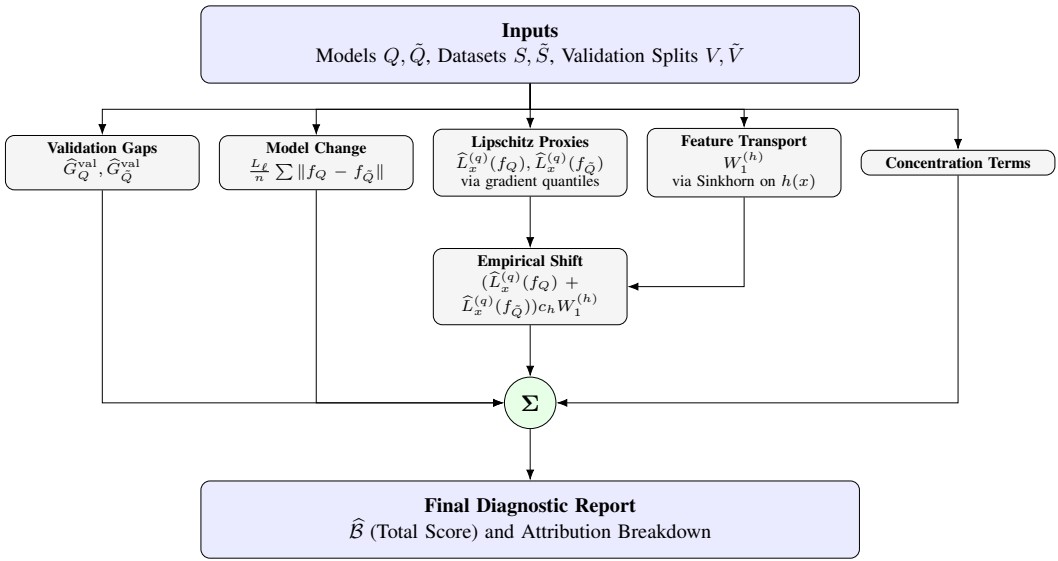

Figure 1: The TRACE diagnostic pipeline. The process takes models and data as input, computes the five core components of the bound in parallel, aggregates them, and produces the final diagnostic score $\widehat{\mathcal{B}}$ and its attribution report.

### 4.6 ALTERNATIVE SHIFT CONTROL VIA MMD

As an alternative to the geometric control offered by Optimal Transport, the COSP term can also be controlled using the functional-analytic framework of Maximum Mean Discrepancy (MMD). This approach bounds the risk change by the product of the MMD between the distributions and the RKHS norm of the model's risk surface. The MMD-based variant is often a lower-variance and more stable estimator, particularly in high-dimensional or low-sample regimes, though it can sometimes result in a more conservative bound. We provide a full, parallel derivation of the MMD-based TRACE diagnostic, including all formal assumptions, propositions, and proofs, in Appendix C. Our implementation includes both the OT and MMD variants, and we recommend reporting the kernel choice, bandwidth, and other relevant hyperparameters when using the MMD diagnostic.

## 5 THEORETICAL EXAMPLE: RIDGE REGRESSION

To build intuition for the TRACE diagnostic and to verify its scaling behavior, we analyze an idealized case of Ridge Regression under a simple covariate shift. We consider a linear predictor $f_w(x) = w^\top x$ with a squared error loss, trained in a population setting where generalization gaps are zero by construction. The data is generated from a true linear relationship $y = w^{*\top} x + \epsilon$. The covariate shift is a simple translation of the input distribution's mean: the source distribution is $P_X = \mathcal{N}(0, I_d)$, and the target is $\tilde{P}_X = \mathcal{N}(\mu, I_d)$. The magnitude of the shift is controlled by $\|\mu\|_2$.

In this setting, the source-trained model $Q$ (with weights $w_Q$) and the target-trained model $\tilde{Q}$ (with weights $w_{\tilde{Q}}$) have different closed-form solutions due to the interaction between the data shift and the L2 regularization (see Appendix E for full derivations). With the generalization gaps removed, the TRACE bound simplifies to its core components: the covariate shift penalty and the model change (algorithmic instability) penalty.

**Analysis of the TRACE Bound Components.** The **covariate shift penalty** is determined by the Wasserstein distance between the two Gaussian distributions, which is simply the Euclidean distance between their means. This term thus scales linearly with the magnitude of the shift:

$$L_x(f_{\tilde{Q}})W_1(P_X, \tilde{P}_X) = L_x(f_{\tilde{Q}})\|\mu\|_2 = O(\|\mu\|_2).$$

The **model change penalty** is controlled by the expected output distance, which depends on the difference in the learned weights, $\Delta w := w_{\tilde{Q}} - w_Q$. As we show in Appendix E, for small shifts, the magnitude of this weight difference is a second-order effect, with $\|\Delta w\|_2 = O(\|\mu\|_2^2)$. Consequently, the model change penalty is of a lower order than the shift penalty in this setting.

**Analysis of the True Risk Difference.** A detailed derivation in Appendix E shows that, for small shifts, the true risk difference scales linearly with the magnitude of the shift, i.e., $|R_P(w_Q) - R_P(w_{\tilde{Q}})| = O(\|\mu\|_2)$.

**Conclusion and Insight.** This idealized example provides a powerful insight. The true risk difference scales linearly with the shift magnitude $\|\mu\|_2$. The dominant term in the TRACE bound, the covariate shift penalty, also scales linearly with $\|\mu\|_2$. This demonstrates that the TRACE inequality is **asymptotically sharp in its scaling**, correctly capturing the rate at which the risk difference grows. While the bound's numerical value may be conservative due to worst-case constants, its components meaningfully reflect the underlying dynamics of the problem.

## 6 EXPERIMENTS

We empirically validate TRACE across a range of benchmarks to answer four key questions: (i) Does the diagnostic correlate with the true risk change $|\Delta R|$? (ii) Is its attribution meaningful? (iii) Is the pipeline practical? and (iv) Can it power an effective deployment gate?

### 6.1 EXPERIMENTAL METHODOLOGY

**Common Setup.** Across all experiments, we evaluate the risk change $|\Delta R|$ on a held-out source test set. The TRACE diagnostic $\widehat{\mathcal{B}}$ is computed using the OT-based instantiation by default. For vision tasks, we use features from a frozen ResNet-50. For a fair comparison of the model change and data shift components, we introduce a label-free calibrated score, **TRACE-Proxy**, used for rank correlation analysis. For deployment gating, we use the uncalibrated, theoretically-grounded **TRACE Gate Score**. Full implementation details, hyperparameter settings, and the formula for TRACE-Proxy are in Appendix F.

**TRACE-Proxy and Attribution.** To address the issue of differing component scales, we use a label-free calibration constant $\hat{c}_h$ to create the TRACE-Proxy score. This constant, estimated once on a small, unlabeled development set, normalizes the scale of the covariate-shift term relative to the model-change term. This ensures the components are comparable in magnitude, enabling the meaningful attribution and ranking analysis that follows.

## 6.2 SYNTHETIC BENCHMARKS: VALIDATION AND ATTRIBUTION

**Goal.** To validate that the TRACE diagnostic tracks $|\Delta R|$ and that its attribution is meaningful in controlled settings.

**Setup and Results.** We applied geometric shifts of increasing severity to two 2D datasets (Gaussian blobs and two moons). To validate individual components, we designed experiments to isolate their effects (e.g., modulating learning rates to isolate model change, or applying geometric transforms to isolate data shift), with full details in Appendix F.6. Across all settings, the TRACE-Proxy score maintained a strong monotonic association with the true risk change (Spearman's $\rho > 0.98$ for blobs, $\rho > 0.75$ for moons). A detailed attribution analysis, provided in Appendix F, confirms that the diagnostic terms behave as expected: the *Empirical Shift* term grows with the geometric severity of the shift, while the *Model Change* term reflects retraining instability. This result confirms the validity and attributional power of TRACE.

## 6.3 CASE STUDY 1: ACTIVE SELECTION ON DOMAINNET

**Goal.** To show how TRACE provides crucial insights in a realistic scenario where data shift and model change interact.

**Setup and Results.** We simulate active domain adaptation, fine-tuning a model trained on `real` images by iteratively selecting and labeling small batches from the `painting` domain. We compare a strategy that minimizes Wasserstein distance (`w1min`) against one that minimizes MMD (`mmdmin`). As detailed in Table 1, TRACE exposes a critical trade-off. While the `mmdmin` strategy successfully reduces the MMD distance, TRACE reveals that the *Model Change* term simultaneously increases, indicating model instability and leading to unreliable performance on the source domain. The `w1min` strategy, in contrast, improves both metrics, leading to consistent gains. The key insight is that monitoring data shift alone is insufficient; TRACE provides a necessary, unified view of both data alignment and model stability.

Table 1: TRACE diagnostics for active selection on DOMAINNET. TRACE reveals that minimizing MMD distance leads to a harmful increase in the **Model Change** (measured by output discrepancy $\mathcal{G}_{\ell_2}$), a trade-off that `w1min` successfully navigates. Columns show: **Labeled** samples used; **Distance** (feature-space alignment); $\mathcal{G}_{\ell_2}$ (model output instability); and $|\Delta R|$ (true source risk degradation).

| Policy | Labeled | Distance ↓ | $\mathcal{G}_{\ell_2}$ ↑ | $|\Delta R|$ |
|---|---|---|---|---|
| `w1min` | 50 | 0.0182 | 42.23 | 5.82 |
| $(W_1)$ | 250 | **0.0034** | **44.68** | **5.11** |
| `mmdmin` | 50 | 0.1772 | 27.77 | 3.35 |
| (MMD) | 250 | **0.0984** | **66.56** | 4.48 |

We stress that this study does not declare `w1min` universally superior. Rather, it demonstrates TRACE's diagnostic value. An engineer monitoring only the **Distance** column would consider `mmdmin` successful as alignment improves. However, TRACE reveals the hidden cost: the **Model Change** term ($\mathcal{G}_{\ell_2}$) simultaneously explodes ($27 \rightarrow 66$), signaling severe instability. This instability drove the poor source performance ($|\Delta R|$). TRACE thus serves as an essential lens, revealing trade-offs invisible to single-metric monitoring.

## 6.4 CASE STUDY 2: DEPLOYMENT GATE ON DOMAINNET

**Goal.** To evaluate the TRACE Gate Score on the high-stakes task of automatically approving or rejecting a candidate model update.

To evaluate the TRACE Gate Score on the high-stakes task of automatically approving or rejecting a candidate model update. This setting mirrors a common MLOps workflow where a new model must pass a "gate" before replacing the current model. Crucially, such decisions often rely on proxy

metrics without fresh labels on the anchor domain. TRACE is designed to plug directly into this workflow, providing an automatic veto signal to prevent harmful updates from being deployed.

**Setup and Results.** We simulate replacing a model trained on `real` images with 20 candidates fine-tuned on the `sketch` domain. Following standard practice, we define an update as **"harmful"** if the true risk on the source test set increases by more than a predefined threshold $\tau$. Formally, an update is harmful if $R_P(\tilde{Q}) - R_P(Q) > \tau$. This frames the problem as a binary classification task for computing AUROC/AUPRC. We use the TRACE Gate Score to predict which updates will be harmful and compare its performance to a standard Maximum Softmax Probability (MSP) baseline (Hendrycks & Gimpel, 2017). The TRACE score is an exceptionally strong predictor of deployment risk, achieving a Spearman's rank correlation with true harm of $\rho \approx \mathbf{0.94}$, significantly outperforming the MSP baseline. As a binary classifier for "harmful" updates, the score yields an AUROC/AUPRC of **1.0** at a key decision threshold, demonstrating its ability to perfectly separate safe from unsafe updates in this setting (see Table 2). This result confirms that TRACE can be operationalized into a reliable, label-efficient safety gate for real-world ML systems.

Table 2: Deployment gate performance. TRACE robustly outperforms the MSP baseline in classifying harmful model updates.

| Method | Rank Correlation ($\rho$) | AUROC (at $\tau = 0.13$) |
|---|---|---|
| TRACE Gate Score | **0.944** | **1.000** |
| MSP Baseline | -0.874 | 0.980 |

The strong discrimination power (AUROC=1.0) and rank correlation ($\rho = 0.94$) validate TRACE's core theoretical property: the bound correctly captures the *relative scaling* of risk change, consistent with the behavior analyzed in Section 5. This structural fidelity enables reliable ordering of candidate models even when exact risk values are unknown.

# 7 CONCLUSION

We introduced TRACE, a framework for attributing the risk change $|\Delta R|$ that occurs when a model is updated on shifted data. TRACE provides a principled, high-probability diagnostic that disentangles this change into interpretable components: generalization error, data shift (measured via OT or MMD), and model change (measured via output distance). Our experiments show that this decomposition is not only theoretically sound but practically effective.

We emphasize that TRACE's diagnostic value derives from its **structural correctness**: the bound scales correctly with shift magnitude (analyzed in Sec. 5) and its components maintain strong monotonic relationships with the true risk change. Our experiments validate this via strong rank correlation ($\rho \approx 0.94$) and near-perfect deployment gate discrimination (AUROC=1.0 in Table 2). This demonstrates that the bound's theoretical structure, which decomposes risk change into interpretable and estimable components, provides the foundation for practical diagnostic and ranking applications in model replacement.

**Limitations and Future Work.** While TRACE provides a powerful diagnostic lens, our current analysis is focused on covariate shift. We view this as both a strength and a limitation. It allows for a complete, end-to-end analysis of one of the most common failure modes in practice (e.g., sensor changes, demographic drift), enabling sharp, actionable diagnostics. However, TRACE does not yet provide formal guarantees for label or concept drift. Extending our two-model attribution perspective to these richer shifts is an important direction for future work. Looking ahead, this work opens the door to **anchor-aware learning**, a new paradigm where the TRACE diagnostic is minimized directly during training to enforce performance guarantees on a chosen anchor domain. This vision connects our attribution framework to future work in certified robustness, conformal prediction for per-update guarantees, and the development of risk-sensitive, multi-anchor safety gates.

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

## A    LLM USAGE

A Large Language Model (LLM) was used as a general-purpose writing and editing assistant in the preparation of this manuscript. Its role was confined to improving the clarity, flow, and structure of the text. The LLM assisted in polishing sentences, suggesting alternative phrasing for conciseness, providing feedback on the organization of sections, and helping to refine the articulation of key concepts. It also provided assistance with LaTeX formatting. All core research ideas, mathematical derivations, experimental design, and scientific claims originated from the human authors, who take full responsibility for the final content of the paper.

## B  PROOFS OF LEMMA 1 AND LEMMA 2

*Proof of Lemma 1.* Insert and subtract $\widehat{R}_S(Q)$, $\widehat{R}_{\tilde{S}}(\tilde{Q})$, and $R_{\tilde{P}}(\tilde{Q})$, then apply the triangle inequality:

$$\left|R_P(Q) - R_P(\tilde{Q})\right| \leq \underbrace{\left|R_P(Q) - \widehat{R}_S(Q)\right|}_{G_Q} + \underbrace{\left|\widehat{R}_S(Q) - \widehat{R}_{\tilde{S}}(\tilde{Q})\right|}_{D_{Q,\tilde{Q}}} + \underbrace{\left|\widehat{R}_{\tilde{S}}(\tilde{Q}) - R_{\tilde{P}}(\tilde{Q})\right|}_{G_{\tilde{Q}}}$$
$$+ \underbrace{\left|R_{\tilde{P}}(\tilde{Q}) - R_P(\tilde{Q})\right|}_{\text{COSP}}$$

**Bounding the population shift term.** Define $\phi_{\tilde{Q}}(x) := \mathbb{E}_{y \sim P(y|x)}[\ell(f_{\tilde{Q}}(x), y)]$. By Assumption 2, $\phi_{\tilde{Q}}$ is $L_x(f_{\tilde{Q}})$-Lipschitz in $x$. Hence, by Kantorovich–Rubinstein duality,

$$\left|R_{\tilde{P}}(\tilde{Q}) - R_P(\tilde{Q})\right| = \left|\mathbb{E}_{x \sim \tilde{P}_X}[\phi_{\tilde{Q}}(x)] - \mathbb{E}_{x \sim P_X}[\phi_{\tilde{Q}}(x)]\right| \leq L_x(f_{\tilde{Q}})\, W_1(P_X, \tilde{P}_X).$$

Combining the two displays yields the claim. $\qquad\qquad\qquad\qquad\qquad\qquad\qquad\qquad\square$

*Proof of Lemma 2.* The goal is to bound the absolute difference between the empirical risks of a fixed model $Q$ on the source sample $S$ and the target sample $\tilde{S}$. The core of the proof is to introduce the expected loss function $\phi_Q(x) := \mathbb{E}_{y \sim P(y|x)}[\ell(f_Q(x), y)]$ as an intermediate bridge. This allows us to separate the effect of the input shift from the statistical noise introduced by the random labels.

**Step 1: Decompose the empirical risk difference.** We add and subtract terms involving $\phi_Q$ to split the total difference into three parts:

$$\widehat{R}_S(Q) - \widehat{R}_{\tilde{S}}(Q) = \left(\frac{1}{n}\sum_{i=1}^n \ell(f_Q(x_i), y_i) - \frac{1}{n}\sum_{i=1}^n \phi_Q(x_i)\right) \quad \text{(Term 1: Source Label Noise)}$$

$$+ \left(\frac{1}{n}\sum_{i=1}^n \phi_Q(x_i) - \frac{1}{n}\sum_{i=1}^n \phi_Q(\tilde{x}_i)\right) \quad\quad \text{(Term 2: Input Shift)}$$

$$+ \left(\frac{1}{n}\sum_{i=1}^n \phi_Q(\tilde{x}_i) - \frac{1}{n}\sum_{i=1}^n \ell(f_Q(\tilde{x}_i), \tilde{y}_i)\right). \quad \text{(Term 3: Target Label Noise)}$$

By the triangle inequality, $|\widehat{R}_S(Q) - \widehat{R}_{\tilde{S}}(Q)| \leq |\text{Term 1}| + |\text{Term 2}| + |\text{Term 3}|$. We now bound each term.

**Step 2: Bound the input shift term (Term 2).** This term represents the difference in the expected loss function $\phi_Q$ evaluated on the source inputs versus the target inputs. Let $\widehat{P}_n$ and $\widehat{\tilde{P}}_n$ be the empirical distributions of the inputs $\{x_i\}$ and $\{\tilde{x}_i\}$, respectively. We can write Term 2 as:

$$\text{Term 2} = \mathbb{E}_{x \sim \widehat{P}_n}[\phi_Q(x)] - \mathbb{E}_{x \sim \widehat{\tilde{P}}_n}[\phi_Q(x)].$$

By Assumption 2, $\ell(f_Q(x), y)$ is $L_x(f_Q)$-Lipschitz in $x$. This property is preserved under expectation, so $\phi_Q(x)$ is also $L_x(f_Q)$-Lipschitz. We can therefore apply Kantorovich-Rubinstein duality to bound the absolute value of this term:

$$|\text{Term 2}| \leq L_x(f_Q)\, W_1(\widehat{P}_n, \widehat{\tilde{P}}_n).$$

**Step 3: Bound the label noise terms (Term 1 and Term 3).** Term 1 is the average of $n$ random variables of the form $Z_i = \ell(f_Q(x_i), y_i) - \phi_Q(x_i)$. For any fixed $x_i$, the expectation of $Z_i$ over the random draw of the label $y_i$ is zero:

$$\mathbb{E}_{y_i \sim P(y|x_i)}[Z_i] = \mathbb{E}_{y_i \sim P(y|x_i)}[\ell(f_Q(x_i), y_i)] - \phi_Q(x_i) = \phi_Q(x_i) - \phi_Q(x_i) = 0.$$

By Assumption 1, the loss $\ell$ is bounded in $[0, M]$, which implies that $\phi_Q(x)$ is also in $[0, M]$. Thus, each $Z_i$ is a zero-mean random variable bounded in $[-M, M]$. The same holds for the random variables in Term 3. We can therefore apply Hoeffding's inequality to both terms.

For a desired total failure probability of $\delta$, we can allocate a budget of $\delta/2$ to each term. By a union bound, we have that with probability at least $1 - \delta$:

$$|\text{Term 1}| + |\text{Term 3}| = \left| \frac{1}{n} \sum_{i=1}^{n} Z_i \right| + \left| \frac{1}{n} \sum_{i=1}^{n} \tilde{Z}_i \right|$$

$$\leq M\sqrt{\frac{1}{2n} \log \frac{4}{\delta}} + M\sqrt{\frac{1}{2n} \log \frac{4}{\delta}} = 2M\sqrt{\frac{1}{2n} \log \frac{4}{\delta}}.$$

**Step 4: Combine the bounds.** Combining the bounds for all three terms via the triangle inequality, we arrive at the final result, which holds with probability at least $1 - \delta$:

$$\left| \widehat{R}_S(Q) - \widehat{R}_{\tilde{S}}(Q) \right| \leq L_x(f_Q) \, W_1(\widehat{P}_n, \widehat{\tilde{P}}_n) + 2M\sqrt{\frac{1}{2n} \log \frac{4}{\delta}}.$$

This completes the proof.

$\square$

## C  FULL DERIVATION OF THE MMD-BASED TRACE DIAGNOSTIC

This section provides the full theoretical and practical details for the Maximum Mean Discrepancy (MMD) based instantiation of the TRACE framework, which serves as an alternative to the Optimal Transport based approach presented in the main text

### C.1  COVARIATE SHIFT CONTROL WITH MMD

The MMD framework replaces the geometric notion of distance with one based on the ability of a class of smooth functions to distinguish between two distributions.

**Maximum Mean Discrepancy.**    We begin by defining the MMD. Let $k(\cdot, \cdot)$ be a positive definite kernel defining a Reproducing Kernel Hilbert Space (RKHS) of functions. In our practical setting, we apply this kernel to feature embeddings $u = h(x)$, inducing a corresponding RKHS on the original input space, which we denote by $\mathcal{H}_h$.

**Definition 2** (MMD). *The Maximum Mean Discrepancy between distributions $\mu$ and $\nu$ is the largest difference in expectations over all functions in the unit ball of the RKHS $\mathcal{H}_h$:*

$$\text{MMD}_{\mathcal{H}_h}(\mu, \nu) := \sup_{\psi \in \mathcal{H}_h, \|\psi\|_{\mathcal{H}_h} \leq 1} |\mathbb{E}_{x \sim \mu}[\psi(x)] - \mathbb{E}_{x \sim \nu}[\psi(x)]|.$$

To connect MMD to the risk change, we require a counterpart to the Lipschitz assumption used in the OT framework. We must assume that our model's risk surface is sufficiently smooth to belong to the RKHS.

**Assumption 4** (RKHS Membership of the Risk Map). *We assume the target risk map $\phi_{\tilde{Q}}(x) := \mathbb{E}_{y \sim P(y|x)}[\ell(f_{\tilde{Q}}(x), y)]$ can be represented by a function in the feature-space RKHS $\mathcal{H}_h$, with a bounded norm denoted $B_{\tilde{Q}}^{(h)}$.*

Under this assumption, the population shift penalty can be bounded by the MMD. The proof is provided in Appendix D.

**Proposition 2** (MMD-based Covariate Shift Control). *Under Assumption 4, the population shift term is bounded by:*

$$|R_P(\tilde{Q}) - R_{\tilde{P}}(\tilde{Q})| \leq B_{\tilde{Q}}^{(h)} \text{MMD}_{\mathcal{H}_h}(P_X, \tilde{P}_X).$$

*Moreover, the population MMD is related to its empirical counterpart, $\widehat{\text{MMD}}$, via standard concentration bounds (Gretton et al., 2012). For any $\delta \in (0, 1)$, with probability at least $1 - \delta$:*

$$\text{MMD}_{\mathcal{H}_h}(P_X, \tilde{P}_X) \leq \widehat{\text{MMD}}_{\mathcal{H}_h}(\widehat{P}_n, \widehat{\tilde{P}}_n) + C_\kappa \sqrt{\frac{1}{n} \log \frac{2}{\delta}},$$

*where $C_\kappa$ is a constant that depends on properties of the kernel $k$.*

Notably, when working in a feature space, the dependence on the feature map $h$ is absorbed into the norm $B_{\tilde{Q}}^{(h)}$, unlike the OT case which requires an explicit scaling factor $c_h$.

## C.2 THE PRACTICAL MMD-BASED TRACE DIAGNOSTIC

To operationalize this bound, we need a method to estimate the RKHS norm $B_{\tilde{Q}}^{(h)}$ and to connect the population MMD to its empirical counterpart.

**Estimating the RKHS Norm via Kernel Ridge Regression.** The theoretical norm $B_{\tilde{Q}}^{(h)}$ of the unknown risk map $\phi_{\tilde{Q}}$ is intractable. We estimate it by learning a proxy function $\widehat{\psi} \in \mathcal{H}_h$ that approximates $\phi_{\tilde{Q}}$ from data. On a held-out set $\{(\tilde{x}_i, \tilde{y}_i)\}_{i=1}^m \sim \tilde{P}$, we create a regression dataset of feature-loss pairs, $\{(u_i, t_i)\}_{i=1}^m$, where $u_i = h(\tilde{x}_i)$ and $t_i = \ell(f_{\tilde{Q}}(\tilde{x}_i), \tilde{y}_i)$. We then fit $\widehat{\psi}$ using Kernel Ridge Regression (KRR):

$$\widehat{\psi} = \arg\min_{\psi \in \mathcal{H}_h} \frac{1}{m} \sum_{i=1}^m (\psi(u_i) - t_i)^2 + \lambda \|\psi\|_{\mathcal{H}_h}^2.$$

By the Representer Theorem, the solution $\widehat{\psi}$ has a known parametric form, $\widehat{\psi}(\cdot) = \sum_{j=1}^m \alpha_j k(u_j, \cdot)$, where the coefficient vector $\boldsymbol{\alpha}$ has a closed-form solution. Let $K$ be the $m \times m$ kernel Gram matrix where $K_{ij} = k(u_i, u_j)$, and let $\mathbf{t}$ be the vector of target losses. The coefficients are given by $\boldsymbol{\alpha} = (K + m\lambda I)^{-1}\mathbf{t}$. The RKHS norm of this solution, $|\widehat{\psi}|_{\mathcal{H}_h} = \sqrt{\boldsymbol{\alpha}^\top K \boldsymbol{\alpha}}$, then serves as our computable estimate, $\widehat{B}_{\tilde{Q}}^{(h)}$. We view this estimate not as a certified upper bound, but as a principled **scale calibrator** for the MMD term. The regularization hyperparameter $\lambda$ is selected via cross-validation.

**Final MMD-based Diagnostic.** By substituting the MMD control from Proposition 2 into the main TRACE decomposition and replacing all population quantities with their empirical estimators, we arrive at the practical MMD-based diagnostic.

**Corollary 2** (Practical MMD-based TRACE). *With probability at least $1 - \delta$, the risk change is bounded by:*

$$|\Delta R| \leq \underbrace{\widehat{G}_Q^{\mathrm{val}} + \widehat{G}_{\tilde{Q}}^{\mathrm{val}}}_{\text{Validation Gaps}} + \underbrace{\widehat{L}_x^{(q)}(f_Q)c_h W_1^{(h)}(\widehat{P}_n, \widehat{\tilde{P}}_n)}_{\text{Empirical Shift (OT)}} + \underbrace{\frac{L_\ell}{n} \sum_{i=1}^n \|f_Q(\tilde{x}_i) - f_{\tilde{Q}}(\tilde{x}_i)\|}_{\text{Model Change}}$$

$$+ \underbrace{\widehat{B}_{\tilde{Q}}^{(h)}\left(\widehat{\mathrm{MMD}}_{\mathcal{H}_h}(\widehat{P}_n, \widehat{\tilde{P}}_n) + C_\kappa\sqrt{\frac{1}{n}\log\frac{2}{\delta}}\right)}_{\text{Population Shift (MMD)}} + \text{Statistical Remainder Terms.}$$

Here, we present a hybrid bound that uses OT for the empirical shift of $Q$ (default) and MMD for the population shift of $\tilde{Q}$. The "Statistical Remainder Terms" consolidate the label noise and validation error terms from Corollary 1.

## C.3 PRACTICAL GUIDANCE FOR THE MMD VARIANT

When implementing the MMD-based diagnostic, several choices are important. For the **kernel**, standard choices like the Gaussian (RBF) or Laplace kernel, applied to normalized features, are recommended. The kernel bandwidth is typically set via the median heuristic. For the **MMD estimator**, unbiased quadratic-time estimators are standard for moderate sample sizes, while linear-time or blocked estimators provide a good trade-off for larger datasets (Gretton et al., 2012). For full reproducibility, all hyperparameters, including the chosen kernel, bandwidth, KRR regularization $\lambda$, and the resulting estimates $\widehat{B}_{\tilde{Q}}^{(h)}$ and $\widehat{\mathrm{MMD}}$, should be reported.

# D  PROOF OF PROPOSITION 2

The proposition consists of two parts. First, we prove the main inequality that bounds the population risk change. Second, we cite the standard concentration result for the MMD estimator.

**Part 1: Bounding the Risk Change.**  Our goal is to bound the population shift term, $|R_P(\tilde{Q}) - R_{\tilde{P}}(\tilde{Q})|$. Under the covariate shift assumption, we can express the risks as expectations of the risk map $\phi_{\tilde{Q}}(x) := \mathbb{E}_{y \sim P(y|x)}[\ell(f_{\tilde{Q}}(x), y)]$ over the input distributions $P_X$ and $\tilde{P}_X$:

$$|R_P(\tilde{Q}) - R_{\tilde{P}}(\tilde{Q})| = |\mathbb{E}_{x \sim P_X}[\phi_{\tilde{Q}}(x)] - \mathbb{E}_{x \sim \tilde{P}_X}[\phi_{\tilde{Q}}(x)]|.$$

The proof relies on two key properties of the Reproducing Kernel Hilbert Space $\mathcal{H}_h$.

First, we use the concept of the **kernel mean embedding**. For any distribution $\mu$, its mean embedding $\mu_\mu \in \mathcal{H}_h$ is a unique element of the RKHS such that for any function $\psi \in \mathcal{H}_h$, the expectation of $\psi$ can be written as an inner product: $\mathbb{E}_{x \sim \mu}[\psi(x)] = \langle \psi, \mu_\mu \rangle_{\mathcal{H}_h}$. Applying this to our risk map $\phi \tilde{Q}$ and distributions $P_X$ and $\tilde{P}_X$, we can rewrite the difference in expectations as:

$$\mathbb{E}_{x \sim P_X}[\phi_{\tilde{Q}}(x)] - \mathbb{E}_{x \sim \tilde{P}_X}[\phi_{\tilde{Q}}(x)] = \langle \phi_{\tilde{Q}}, \mu_{P_X} \rangle_{\mathcal{H}_h} - \langle \phi_{\tilde{Q}}, \mu_{\tilde{P}_X} \rangle_{\mathcal{H}_h} = \langle \phi_{\tilde{Q}}, \mu_{P_X} - \mu_{\tilde{P}_X} \rangle_{\mathcal{H}_h}.$$

Second, we take the absolute value and apply the Cauchy-Schwarz inequality, which states that $|\langle a, b \rangle| \le |a||b|$:

$$\left| \langle \phi_{\tilde{Q}}, \mu_{P_X} - \mu_{\tilde{P}_X} \rangle_{\mathcal{H}} \right| \le \|\phi_{\tilde{Q}}\|_{\mathcal{H}} \|\mu_{P_X} - \mu_{\tilde{P}_X}\|_{\mathcal{H}}.$$

By definition, the MMD is the distance between the kernel mean embeddings: $\mathrm{MMD}_{\mathcal{H}_h}(P_X, \tilde{P}_X) = \|\mu_{P_X} - \mu_{\tilde{P}_X}\|_{\mathcal{H}_h}$. Furthermore, by Assumption 4, we have $\|\phi_{\tilde{Q}}\|_{\mathcal{H}_h} \le B_{\tilde{Q}}^{(h)}$. Substituting these into the inequality yields the final result:

$$|R_P(\tilde{Q}) - R_{\tilde{P}}(\tilde{Q})| \le B_{\tilde{Q}}^{(h)} \mathrm{MMD}_{\mathcal{H}_h}(P_X, \tilde{P}_X).$$

**Part 2: Concentration of the MMD Estimator.**  The second statement in the proposition is a standard concentration inequality for the empirical MMD estimator. It bounds the deviation of the empirical MMD, computed on finite samples, from the true population MMD. The proof and a detailed analysis of the constants can be found in the foundational work on kernel two-sample tests, for instance, in Theorem 12 of Gretton et al. (2012). The result states that for any $\delta \in (0, 1)$, with probability at least $1 - \delta$:

$$\mathrm{MMD}_{\mathcal{H}_h}(P_X, \tilde{P}_X) \le \widehat{\mathrm{MMD}}_{\mathcal{H}_h}(\hat{P}_n, \widehat{\tilde{P}}_n) + C_\kappa \sqrt{\frac{1}{n} \log \frac{2}{\delta}},$$

This completes the justification for the proposition.

# E  DERIVATIONS FOR THE RIDGE REGRESSION EXAMPLE

Here we provide the detailed algebraic derivations for the theoretical example in Section 5.

**Closed-form Solutions for Weights.**  We analyze the models in a **population setting**, where they are learned by minimizing the true expected risk. The model $Q$, trained on source data, and the model $\tilde{Q}$, trained on target data, have weights

$$w_Q = \arg \min_w \mathbb{E}_{x \sim P, \epsilon}\left[ (w^\top x - y)^2 + \lambda \|w\|_2^2 \right], \tag{8}$$

$$w_{\tilde{Q}} = \arg \min_w \mathbb{E}_{x \sim \tilde{P}, \epsilon}\left[ (w^\top x - y)^2 + \lambda \|w\|_2^2 \right]. \tag{9}$$

The well-known solutions are (Bishop & Nasrabadi, 2006, Chapter 3):

$$w_Q = (I + \lambda I)^{-1} \mathbb{E}_P[xx^\top] w^* = \frac{1}{1 + \lambda} w^*, \tag{10}$$

$$w_{\tilde{Q}} = \left( \mathbb{E}_{\tilde{P}}[xx^\top] + \lambda I \right)^{-1} \mathbb{E}_{\tilde{P}}[xx^\top] w^* \tag{11}$$

$$= (I + \mu\mu^\top + \lambda I)^{-1}(I + \mu\mu^\top) w^*. \tag{12}$$

Critically, $w_Q \ne w_{\tilde{Q}}$. The shift in the data distribution's first moment ($\mu$) interacts with the regularization ($\lambda$) to produce a different optimal model, instantiating the concept of algorithmic instability.

**Derivation of the Weight Difference** $\Delta w$. The **algorithmic instability penalty** is given by the expected output distance on the target distribution:

$$L_\ell \cdot \mathbb{E}_{x \sim \tilde{P}} |f_{w_Q}(x) - f_{w_{\tilde{Q}}}(x)| = L_\ell \cdot \mathbb{E}_{x \sim \tilde{P}} |(w_Q - w_{\tilde{Q}})^\top x|. \tag{13}$$

Letting $\Delta w := w_{\tilde{Q}} - w_Q$, one can use the Sherman–Morrison formula to find its exact form:

$$\Delta w = \frac{\lambda(\mu^\top w^*)}{(1 + \lambda)(1 + \lambda + \|\mu\|^2)} \, \mu. \tag{14}$$

The expectation term in equation 13 can be further bounded using the Cauchy–Schwarz inequality as

$$\mathbb{E}_{x \sim \tilde{P}} \left[ |\Delta w^\top x| \right] \leq \|\Delta w\|_2 \, \mathbb{E}_{x \sim \tilde{P}} \|x\|_2. \tag{15}$$

Here, $\mathbb{E}_{x \sim \tilde{P}} \|x\|_2$ is $O(1)$ (constant for fixed dimension), and

$$\|\Delta w\|_2 = |c| \, \|\mu\|, \quad \text{where} \quad c = \frac{\lambda(\mu^\top w^*)}{(1 + \lambda)(1 + \lambda + \|\mu\|^2)}. \tag{16}$$

For small $\|\mu\|$, we can expand the denominator using a Taylor series:

$$\frac{1}{1 + \lambda + \|\mu\|^2} = \frac{1}{1 + \lambda} \left( 1 - \frac{\|\mu\|^2}{1 + \lambda} + O(\|\mu\|^4) \right), \tag{17}$$

which gives

$$c = \frac{\lambda(\mu^\top w^*)}{(1 + \lambda)^2} + O(\|\mu\|^3) = O(\|\mu\|). \tag{18}$$

Therefore, $\|\Delta w\|_2 = |c| \, \|\mu\| = O(\|\mu\|^2)$. Consequently, the instability penalty is of lower order than the shift penalty in this setting.

**Derivation of the True Risk Difference Scaling.** The true risk on the source distribution for a model with weights $w$ is

$$R_P(w) = \mathbb{E}_P[(w^\top x - y)^2] = \|w - w^*\|^2 + \sigma^2. \tag{19}$$

The quantity of interest on the LHS of the TRACE bound is therefore

$$\text{LHS} = |R_P(w_Q) - R_P(w_{\tilde{Q}})| = \left| \|w_Q - w^*\|^2 - \|w_{\tilde{Q}} - w^*\|^2 \right|. \tag{20}$$

Using the vector identity $\|a\|^2 - \|b\|^2 = (a - b)^\top (a + b)$, we get

$$\text{LHS} = |(w_Q - w_{\tilde{Q}})^\top (w_Q + w_{\tilde{Q}} - 2w^*)| = |(-\Delta w)^\top (2(w_Q - w^*) + \Delta w)|. \tag{21}$$

The first factor, $\Delta w$, scales linearly with the shift, i.e., $O(\|\mu\|)$. The second factor,

$$2(w_Q - w^*) + \Delta w, \tag{22}$$

converges to the constant vector $2(w_Q - w^*)$ as $\mu \to 0$. Therefore, for small shifts, the LHS scales as $O(\|\mu\|)$.

# F    ADDITIONAL EXPERIMENTAL DETAILS

This section provides the full, detailed setup for the experiments described in Section 6.

## F.1    ANCHOR DOMAIN, RISK-CHANGE SIGN, AND EVALUATION PROTOCOL

We fix a source/anchor distribution $P$ and consider replacing $Q$ with $\tilde{Q}$ under covariate shift ($P(y \mid x){=}\tilde{P}(y \mid \tilde{x})$, $P_X \neq \tilde{P}_X$). We define the *signed* risk change

$$\Delta R = R_P(Q) - R_P(\tilde{Q}),$$

so $\Delta R {>} 0$ indicates an improvement on the anchor, and $\Delta R {<} 0$ indicates harm. Unless explicitly stated, all plots and rank-power metrics in the main text use the *absolute* change $|\Delta R|$ to avoid sign confounds. We still compute the signed $\Delta R$ for gate labeling (harmful $\Leftrightarrow \Delta R {>} \tau$) and for any analysis where directionality is required.

**What we report by default.** (i) the true absolute change $|\Delta R|$ on a held-out source test set; (ii) rank-power via a calibrated, label-free score (*TRACE-Proxy*; Sec. F.3); (iii) an attribution breakdown from the full diagnostic $\widehat{\mathcal{B}}$.

## F.2 Common Implementation Details

**Transport ($W_1^{(h)}$).** We use the *debiased* Sinkhorn divergence in feature space $h(x)$ with squared Euclidean ground cost ($p{=}2$), $\varepsilon{=}0.1$, 200 iterations.

**Sinkhorn Divergence Computation.** To provide further clarity, the Sinkhorn divergence is an entropy-regularized version of the Optimal Transport distance. It is computationally much faster than solving the exact linear programming OT problem, especially when parallelized on GPUs. While entropic regularization introduces a bias compared to the true Wasserstein distance, we use the *debiased* variant, which subtracts the self-transport terms to yield a more stable and accurate approximation of the true $W_1$ distance.

**Sinkhorn Divergence Scalability.** In practice, GPU-accelerated libraries (such as POT or Geom-Loss (Feydy et al., 2019)) allow computation on tens of thousands of samples in seconds. For very large datasets (e.g., $> 50$k samples), we employ standard subsampling to compute the distance on mini-batches. Since we compute distances on feature embeddings (typically 512–2048 dimensions), the method is efficient even for large models, ensuring TRACE remains scalable to industrial-sized benchmarks.

**Feature map $h$.** Frozen ResNet-50 (ImageNet) penultimate features; per-sample $\ell_2$ normalization; no whitening.

**Lipschitz proxies.** For model $f$ we estimate the $q$-quantile gradient norm $\widehat{L}_x^{(q)} := \mathrm{Quantile}_q\big(\|\nabla_x \ell(f(x), y)\|\big)$ on held-out splits ($V$ for $Q$, $\tilde{V}$ for $\tilde{Q}$). Unless stated, $q{=}0.99$. Dvoretzky–Kiefer–Wolfowitz (DKW) provides nonparametric CIs for quantiles.

**Output discrepancy (model change).** On the same *target* inputs $\tilde{S}$,

$$\mathcal{G}_{\ell_2} = \frac{1}{n}\sum_{i=1}^{n} \big\| f_Q(\tilde{x}_i) - f_{\tilde{Q}}(\tilde{x}_i) \big\|_2.$$

**Loss scale and clipping.** Cross-entropy with logits clipped to $[-10, 10]$, so $\ell \in [0, M]$ with $M{=}10 + \log C$. Clipping stabilizes gradient-quantile estimates and makes the empirical diagnostic numerically well-scaled.

**Splits and seeds.** Stratified $15\%$ validation on both $S$ and $\tilde{S}$; 4 seeds $\{0, 1, 2, 3\}$. Bootstrap $95\%$ CIs are taken over seeds unless noted.

## F.3 Diagnostic vs. Ranking Score, and Calibration

**Diagnostic (*TRACE*).** All bound/tightness claims use the full diagnostic $\widehat{\mathcal{B}}$.

**Ranking score (*TRACE-Proxy*).** Used for correlation/power analyses,

$$\text{TRACE-Proxy} = \underbrace{\mathcal{G}_{\ell_2}}_{\text{model change}} + \widehat{c}_h \underbrace{\widehat{L}_x^{(q)} \cdot \text{Distance}}_{\text{covariate shift}}, \qquad \text{Distance} \in \{W_1^{(h)}, \text{MMD}\}. \quad (23)$$

**Calibration of $\widehat{c}_h$ (unsupervised).** For each (source,target) pair we choose a small development set of 3–5 candidates (disjoint from evaluation) and set

$$\widehat{c}_h = \frac{\mathrm{median}(\mathcal{G}_{\ell_2})}{\mathrm{median}\big(\widehat{L}_x^{(q)} \cdot \text{Distance}\big)}, \quad (24)$$

then *freeze* $\widehat{c}_h$ for all subsequent results.

**Interpretation of $\widehat{c}_h$.** The computable TRACE bound combines (i) two input-Lipschitz factors, (ii) a representation scale factor $c_h$ for $h$, and (iii) a distance proxy for the population COSP term.

TRACE-Proxy collapses these into a single $\widehat{L}_x^{(q)} \cdot \text{Distance}$; $\widehat{c}_h$ absorbs (a) feature-scale $c_h$, (b) the two→one Lipschitz collapse, and (c) normalization between empirical and population distances. *Only* the proxy uses $\widehat{c}_h$; the gate scores below retain explicit factors.

**Synthetic note.** In synthetic experiments we also report the full $\widehat{\mathcal{B}}$ alongside TRACE-Proxy; proxy calibration is still performed once and kept fixed.

### F.4 GATE SCORES FOR DEPLOYMENT

We use two fixed variants; higher values predict more harm on the anchor domain:

$$\text{TRACE-W} := \mathcal{G}_{\ell_2} \;+\; L_x\Big(f_{\tilde{Q}}\Big)\, W_1^{(h)}(\widehat{P}_n, \widehat{\tilde{P}}_n), \tag{25}$$

$$\text{TRACE-MMD} := \mathcal{G}_{\ell_2} \;+\; B_{\tilde{Q}}^{(h)}\, \text{MMD}_{\mathcal{H}_h}(P_X, \tilde{P}_X). \tag{26}$$

Generalization-gap terms are omitted for stable fine-tuning (omission does not affect ranking in our settings). When the distance term is (nearly) constant across candidates, both scores reduce to $\mathcal{G}_{\ell_2}$.

### F.5 METRICS, TIGHTNESS, AND REPORTING CONVENTIONS

**Rank-power.** Spearman's $\rho$ between TRACE-Proxy and $|\Delta R|$.

**Tightness.** We report the dimensionless ratio $\widehat{\mathcal{B}}/|\Delta R|$ whenever $\Delta R \neq 0$.

**Reproducibility keys.** Frozen $h$; Sinkhorn $\varepsilon=0.1$, 200 iters; `eval`-mode logits clipped to $[-10, 10]$ for $\mathcal{G}_{\ell_2}$; $q=0.99$ gradient-quantile proxies (GT labels when available; otherwise $Q$'s pseudo-labels). Seeds/splits as in Sec. F.2. Unsupervised $\widehat{c}_h$ calibration (Eq. 24) once per pair.

### F.6 SYNTHETIC BENCHMARKS: WORLDS, PROTOCOL, AND RESULTS

**Worlds.** Two label-preserving shifts:

- **Blobs-Shift (Logistic/MLP):** two Gaussian blobs with rotation+translation; linear and shallow-MLP variants.
- **Moons-Warp (Nonlinear MLP):** `make_moons` with an invertible radial twist; $h(x)$ is taken from the penultimate layer.

**Protocol.** Sweep shift magnitude, sample sizes $n, \tilde{n} \in \{1000, 2000, 5000\}$, and regularization. For each configuration we compute $|\Delta R|$ on a large source test set ($N \geq 10^5$), the full diagnostic $\widehat{\mathcal{B}}$, and TRACE-Proxy (Eq. 23). $\widehat{c}_h$ is calibrated once and frozen.

**Validating Individual TRACE Components.** To validate the attributional power of individual TRACE components, we designed controlled synthetic experiments that isolate specific factors:

- **Isolating Model Change:** We fix the data shift and fine-tune models with varying hyperparameters (e.g., learning rates ranging from $10^{-4}$ to $10^{-2}$). This keeps the data shift term constant while modulating the *Model Change* term ($\mathcal{G}_{\ell_2}$). We verify that increases in this term correlate with a rise in $|\Delta R|$, confirming its role in diagnosing model instability.
- **Isolating Data Shift:** We fix the training procedure and sweep the geometric shift magnitude (e.g., translation distance). This directly modulates the data shift term ($W_1^{(h)}$) while keeping algorithmic parameters constant. We confirm that the TRACE diagnostic correctly tracks the corresponding increase in risk, validating the role of the COSP term.

These controlled experiments confirm that the components of TRACE successfully isolate and quantify their corresponding sources of risk.

**Main results (rank-power).** Spearman's $\rho$ aggregated over sweeps: Blobs—OT: 0.984, MMD: 0.984; Moons—OT: 0.818, MMD: 0.758.

**Attribution behavior (Interpreting Table 3).** In the *Blobs-Shift* scenario, the *ModelChange* term dominates at mild severities, while both terms grow under stronger shifts. For *Moons-Warp*, we walk through specific rows to show how the diagnostic disentangles risk sources:

- **Mild Shift ($\alpha$=0.25):** The risk change is minimal ($|\Delta R| \approx 0.01$). The diagnostic $\widehat{\mathcal{B}}$ remains low ($\approx$ 0.8). Crucially, comparing to the baseline ($\alpha$=0), the *ModelChange* term remains stable ($\approx 0.48$ vs. 0.50 at baseline), while the *EmpShift* term rises (from 0.06 to 0.32) to reflect the geometric change. This shows TRACE does not raise false alarms for benign shifts.
- **Severe Shift ($\alpha$=2.0):** The risk degrades significantly ($|\Delta R| \approx 0.33$). The diagnostic scales correctly ($\widehat{\mathcal{B}} \approx 21.5$), with the *ModelChange* term spiking to 12.0 alongside *EmpShift* (9.5). This signals that the model has become unstable under severe distribution shift—the update is genuinely unsafe.

This demonstrates that TRACE's attribution is regime-dependent: it distinguishes benign geometric shifts from genuine model failures, enabling safe deployment decisions.

Table 3: Representative synthetic runs: monotonic growth of $\widehat{\mathcal{B}}$ with shift severity and per-term contributions; rows vary $G_Q, G_{\bar{Q}}$ via $n$ and severity.

| World | Shift | $n$ | $|\Delta R|$ | $\widehat{\mathcal{B}}_{\text{OT}}$ | $\widehat{\mathcal{B}}_{\text{MMD}}$ | EmpShift$_{\text{OT}}$ | EmpShift$_{\text{MMD}}$ | ModelChange | $G_Q/G_{\bar{Q}}$ |
|---|---|---|---|---|---|---|---|---|---|
| *Blobs-Shift* (rot= 0°) | | | | | | | | | |
| tr=0.25 | 0.25 | 1000 | 0.0470 | 1.73 | 1.73 | 0.0008 | 0.0042 | 1.7240 | 0.0033 / 0.0034 |
| tr=0.25 | 0.25 | 2000 | 0.0468 | 2.09 | 2.10 | 0.0003 | 0.0018 | 2.0648 | 0.0148 / 0.0147 |
| tr=1.0 | 1.00 | 2000 | 0.7097 | 32.72 | 32.72 | 0.0047 | 0.0084 | 32.6832 | 0.0148 / 0.0147 |
| *Moons-Warp* | | | | | | | | | |
| $\alpha$=0.0 | 0.0 | 1000 | 0.0166 | 0.57 | 0.50 | 0.0628 | 0.0000 | 0.5004 | 0.0016 / 0.0013 |
| $\alpha$=0.25 | 0.25 | 2000 | 0.0115 | 0.81 | 1.01 | 0.3175 | 0.5174 | 0.4815 | 0.0113 / 0.0026 |
| $\alpha$=1.0 | 1.0 | 2000 | 0.0490 | 6.41 | 5.94 | 3.3639 | 2.8969 | 3.0276 | 0.0125 / 0.0017 |
| $\alpha$=2.0 | 2.0 | 2000 | 0.3302 | 21.54 | 17.73 | 9.5040 | 5.6925 | 12.0233 | 0.0133 / 0.0019 |

**Why Spearman stays high.** Under our constructions, both $W_1^{(h)}$ and RBF-MMD increase with geometric severity; adding $\mathcal{G}_{\ell_2}$ on the same inputs preserves rank ordering across runs, keeping $\widehat{\mathcal{B}}$ monotone in $|\Delta R|$ even as absolute scale depends on Lipschitz proxies and regularization.

F.7 CASE STUDY: ACTIVE SELECTION ON DOMAINNET

**Scenario.** Adapt $Q$ trained on `real` to `painting` by iteratively selecting and labeling small batches.

**Selection policies.** At round $t$, choose batch $\mathcal{B}_t$ of size $b$=50 by minimizing a source–target discrepancy in $h$:

1. **w1min**: Distance $= W_1^{(h)}$ (squared Euclidean cost).
2. **mmdmin**: Distance $=$ MMD (RBF kernel; median bandwidth on $h$).

Formally, $\mathcal{B}_t = \arg\min_{|\mathcal{B}|=b} \text{Distance}\big(h(S_{\text{src}}), h(\mathcal{B})\big)$, recomputed each round.

**Scoring.** Ranking uses *TRACE-Proxy* with frozen $\widehat{c}_h$; attribution uses $\widehat{\mathcal{B}}$.

**Protocol.** Experiments use **high-dimensional (2048-d)** feature embeddings extracted from the frozen ResNet-50 backbone. Target pool $\approx$50k `painting` images; rounds add 50 labels each (totals $50, 100, 150, 200, 250$). Fine-tuning: SGD (lr $3 \times 10^{-4}$, momentum 0.9), 5 epochs/round; same augmentations as source; early stop on a target validation split.

**Findings.** w1min: $W_1^{(h)}$ shrinks; $\mathcal{G}_{\ell_2}$ shows a small early bump then declines; $|\Delta R|$ improves monotonically. mmdmin: MMD falls but $\mathcal{G}_{\ell_2}$ rises sharply, making anchor improvements unreli-

able. TRACE reveals that monitoring *only* alignment can be misleading; model-change must be controlled.

Table 4: TRACE diagnostics on DOMAINNET (`source/anchor`: `real`, `target`: `painting`). Each round re-trains $\tilde{Q}$. $|\Delta R|$ is in p.p. of top-1 error on `real`. The bound ratio ($\widehat{\mathcal{B}}/|\Delta R|$) remains stable (5–9), confirming that the diagnostic tracks the scale of the true risk consistent with our theoretical analysis.

| Policy | Round | Labeled | Distance | Out. Disc. ($\ell_2$) | TRACE-Proxy | $|\Delta R|$ | $\widehat{\mathcal{B}}/|\Delta R|$ |
|---|---|---|---|---|---|---|---|
| | | | | **w1min (Distance $= W_1^{(h)}$)** | | | |
| | 1 | 50 | 0.0182 | 42.23 | 42.74 | 5.82 | 7.34 |
| | 2 | 100 | 0.0051 | 48.94 | 49.11 | 8.71 | 5.64 |
| | 3 | 150 | 0.0043 | 47.02 | 47.17 | 6.76 | 6.98 |
| | 4 | 200 | 0.0036 | 45.42 | 45.55 | 5.71 | 7.98 |
| | 5 | 250 | 0.0034 | 44.68 | 44.79 | 5.11 | 8.76 |
| | | | | **mmdmin (Distance $=$ MMD)** | | | |
| | 1 | 50 | 0.1772 | 27.77 | 39.73 | 3.35 | 11.86 |
| | 2 | 100 | 0.1287 | 39.09 | 51.09 | 3.69 | 13.85 |
| | 3 | 150 | 0.1062 | 54.95 | 65.85 | 2.21 | 29.80 |
| | 4 | 200 | 0.0994 | 53.49 | 65.45 | 4.40 | 14.88 |
| | 5 | 250 | 0.0984 | 66.56 | 79.89 | 4.48 | 17.83 |

### F.8 CASE STUDY: DEPLOYMENT GATE ON DOMAINNET (REAL $\rightarrow$ SKETCH)

**Setup.** Evaluate 20 fine-tuned candidates (`sketch`) against $Q$ (`real`) on the anchor $P=$`real`. Baseline: `-MSP` (negated maximum softmax probability). We utilize the same **high-dimensional (2048-d)** feature space as the active selection task to ensure a realistic evaluation of high-stakes visual shifts.

**Metrics.** (i) *Ranking*: Spearman's $\rho$ vs. $|\Delta R|$. (ii) *Gating*: positive=harmful ($\Delta R > \tau$); AUROC/AUPRC across thresholds.

**Results.** TRACE correlates strongly with harm ($\rho \approx 0.944$), outperforming `-MSP` ($\rho \approx -0.874$). In this fine-tuning setting, both $W_1^{(h)}$ and single-bandwidth MMD are nearly constant across candidates, making the gate effectively $\mathcal{G}_{\ell_2}$. A threshold sweep shows robust discrimination; at $\tau=0.13$ (55% harmful), TRACE attains AUROC/AUPRC = 1.000.

Table 5: Threshold sweep for the deployment gate (positive = harmful). TRACE remains robust across operating points.

| $\tau$ | AUROC_TRACE | AUPRC_TRACE | AUROC_-MSP | AUPRC_-MSP |
|---|---|---|---|---|
| 0.10 | 1.000 | 1.000 | 1.000 | 1.000 |
| 0.13 | **1.000** | **1.000** | 0.980 | 0.986 |
| 0.23 | 0.984 | 0.950 | 0.984 | 0.950 |

### F.9 REFERENCE POLICY

Across all sections and tables:

- "TRACE-Proxy" means Eq. equation 23 with $\widehat{c}_h$ calibrated by Eq. equation 24.

- "TRACE" or "the diagnostic" means the full $\widehat{\mathcal{B}}$.

- Gate scores are *TRACE-W/TRACE-MMD* as in Eqs. equation 25–equation 26.

