# OpenReview forum: "TRACE: Theoretical Risk Attribution under Covariate-shift Effects"
_ICLR.cc/2026/Conference — Submitted to ICLR 2026_

### Official Review · Reviewer_Tg6A · 2025-10-30

**Soundness:** 3
**Presentation:** 2
**Contribution:** 3
**Rating:** 6
**Confidence:** 4

**Summary:**

The authors propose an interpretable analytic upper bound for a predictive model on the difference between the risk of the source model and the risk of the adapted model on the source domain. The authors identify four factors constructing the bound: generalization gap terms, empirical risk gap, and the population risk gap, and then provide a computable proxy version of the bound that replaces all population terms with empirically estimatable ones. Finally, they provide a conceptual diagnostic framework by utilizing the components of the bound to attribute model's failure.

**Strengths:**

* [`Exceptional Clarity`] The derivation of the proposed bound is easy to follow. No unnecessary complex tweak. The flow of paper is very straightforward.
* [`Practical Bound`] The proposed bound captures multiple sources of the model's failure mode, and those sources are clearly decomposed.
* [`User-friendly Theoretical Framework`] Not only providing just an analytic form of bound, but the authors explicitly instantiate all the individual components of the proposed bound, so that one can easily leverage it.
* [`Appropriate Descriptions on Application and Example`] Besides, the authors give a diagnostic pipeline to attribute the model's failure and provide minimal working examples (Ridge Regression on a synthetic dataset and a computer vision model on a real image dataset).

**Weaknesses:**

* [`Realism of Assumptions`] The authors made some assumptions about the loss -- boundness and Lipschitz property, but did not discuss how it is in case in practice.
* [`Discussion on the Bound Tightness`] Although I know that it is inevitable to introduce multiple inequalities to make the bound empirically computable, I believe that a discussion on how the tightness of the bound is affected by each empirical approximation and when it can achieve the tightest/loosest bound should be included.
* [`Limited Coverage of Theoretical Framework`] The proposed framework was built on the C-way classification model. There is no discussion about how the proposed bound can be extended to a broader class of modern models.
* [`Limited Effectiveness of Application`] As the scales of each component in the bound may differ, we don't actually know which part matters than the other parts relatively. How can we genuinely attribute the failure of the model, and how precise is it?

**Questions:**

> Minor suggestions
* L107: I think you misuse the (") in this sentence.
* In the abstract L013, when you introduce $\Delta R:= R_{P}(Q)-R_{P}(\tilde{Q})$, how about denoting it $\Delta R:= R_{P}(\tilde{Q})-R_{P}(Q)$ to make it more intuitive (even though it does not matter because only the $|\Delta R|$ will be used)?
* Table 1 is hard to read. There are no clear descriptions of the meaning of each column.
* Could you add an explanation of the Sinkhorn distance computation for further clarity?

---

If there is any misunderstanding from me, please point it out.

---

> ### Author Response · Authors · 2025-11-21
> **Response to Reviewer Tg6A**
>
> We thank the reviewer for their strong endorsement of our paper’s clarity and utility, noting our derivation is "easy to follow" and the framework is "user-friendly" and "practical." We appreciate the constructive feedback on our assumptions and analysis. We address the specific points below.
>
> **1. On the Realism of Assumptions (Weakness 1).**
>
> These are standard and practical assumptions. We will add an expanded discussion to the paper.
> *   **Boundedness (Ass. 1):** This is enforced by clipping logits (e.g., to [-10, 10]), a standard technique for numerical stability that ensures strict boundedness without significantly affecting model performance.
> *   **Lipschitz Continuity (Ass. 2 & 3):** These are realistic for modern networks. For the standard softmax cross-entropy loss, the function is globally Lipschitz w.r.t. its logit inputs with a known constant $L_\ell \le \sqrt{2}$. For inputs, Lipschitz continuity is naturally encouraged by standard regularization like weight decay. Furthermore, our method's strength is that we do not rely on opaque theoretical constants; we estimate the effective Lipschitzness using data-driven proxies (gradient quantiles), ensuring the assumption reflects the actual trained model.
>
> **2. On Bound Tightness, Scales, and Practical Attribution (Weaknesses 2 & 4).**
>
> This is a crucial question: how can we perform meaningful attribution if bound components have different scales or looseness?
> *   **Relative Tightness:** We prioritize scaling correctness over absolute tightness. Our Ridge Regression analysis proves the bound is **asymptotically sharp** in its scaling with shift magnitude. Empirically, this translates to high ranking fidelity ($\rho \approx 0.94$), meaning the bound is "tight enough" to reliably distinguish safe from unsafe updates.
> *   **Calibrated Attribution via TRACE-Proxy:** You are correct that raw terms may have different scales. The calibration constant $\hat{c}_h$ in our TRACE-Proxy, estimated *without labels*, explicitly **normalizes the scale of the covariate-shift term relative to the model-change term**. This balancing allows them to be combined into a single, meaningful diagnostic, directly enabling the practical attribution shown in the experiments. We will clarify this critical role of unsupervised calibration in Section 6.
>
> **3. On the Generality of the Theoretical Framework (Weakness 3).**
>
> Thank you for the opportunity to clarify this. The TRACE framework is **not limited to classification**. The derivation holds for any bounded, Lipschitz loss function. Indeed, our theoretical analysis in **Section 5 specifically covers Ridge Regression**, proving its applicability to continuous regression tasks. While our main experiments focus on classification for standard benchmarking, the framework is model-agnostic. We will revise the problem setup to explicitly state this generality.
>
> **4. Responses to Minor Suggestions.**
> We appreciate the careful reading and will incorporate all suggestions in the revised manuscript:
> *   We will correct the typo (L107) and adjust the abstract notation for clarity.
> *   **Definition of $\Delta R$:** We chose the convention $\Delta R = R_P(Q) - R_P(\tilde{Q})$ so that a positive value represents an **improvement** (reduction) in risk. To ensure clarity, we have added a footnote in Section 3 explicitly stating this sign convention.
> *   We will improve the caption of Table 1 to explicitly define all columns.
> *   We will add a paragraph to the appendix detailing our use of the **debiased Sinkhorn divergence**, which corrects for entropic bias, and its scalable computation via subsampling and GPU acceleration.

---

> ### Comment · Reviewer_Tg6A · 2025-11-21
>
> I really appreciate the authors' professional rebuttal!
>
> My initial concerns on assumption realism, bound tightness, attribution in practice, and framework generality are somewhat addressed.
> Besides, the authors significantly improved their presentation during this rebuttal, so I raised the presentation rating from 2 to 3 accordingly.
>
> ---
>
> Given that all other reviewers are less confident than I am,
> I think I should make a bold recommendation: "8: accept, good paper (poster)".
> To be honest, I don't think this work made some ground-breaking contributions; However, there are many theoretical insights, takeaways, and practical implications that are sufficient to recommend this paper towards acceptance.
>
> ---
>
> One optional suggestion is to try to easily explain your awesome theoretical framework through some intuitive visualization, as done in Figure 1 and Figure 2 of [Anonymous OpenReview 2025].
> As a reviewer who has some background in this field, I took an intensive look at the paper and was able to understand the great potential of this paper. However, there can be general/novice readers who are less familiar with this topic and take a bird's-eye view of the paper. If the authors want to catch their interests as well, I would recommend adding some intuitive visualizations of what you do.
>
> 1. Anonymous OpenReview 2025, "How Do Transformers Learn to Associate Tokens: Gradient Leading Terms Bring Mechanistic Interpretability"
>
> ---
>
> Anyway, good luck authors!

---

> > ### Author Response · Authors · 2025-11-25
> > **Response to Reviewer Tg6A**
> >
> > Dear Reviewer Tg6A,
> >
> > We are incredibly grateful for your continued engagement and your strong endorsement of our work. We sincerely thank you for raising your rating and for your "bold recommendation" to accept.
> >
> > We are especially encouraged that you found our rebuttal effective in addressing your concerns regarding assumption realism and bound tightness.
> >
> > **Regarding the Intuitive Visualization:**
> > This is a fantastic suggestion. We agree that a conceptual "bird's-eye view" visualization would significantly improve accessibility for readers less familiar with the topic. We commit to designing and including this figure in the final camera-ready version of the paper.
> >
> > Thank you again for your time, your detailed feedback, and for championing our paper. Your review has significantly strengthened the final manuscript.
> >
> > Best regards,
> > The Authors

---

### Official Review · Reviewer_ns7s · 2025-10-31

**Soundness:** 3
**Presentation:** 2
**Contribution:** 2
**Rating:** 4
**Confidence:** 2

**Summary:**

This paper introduces TRACE standing for Theoretical Risk Attribution under Covariate shift Effect. The key idea is to give a full diagnosis of the change in the risk when a model that was trained on a source distribution is deployed on a different target distribution. More precisely TRACE decomposes the error as the sum of 4 terms: the source generalisation gap, the target generalisation gap, the empirical discrepancy and equivalent shift penalty term under a set of hypothesis explicit expression for each of the term are given the paper then include some experiment on two dimensional synthetic dataset and DomainNet.

**Strengths:**

- The paper is well written and addresses important question for transfer learning
- The paper describes the entire pipeline to compute the estimate

**Weaknesses:**

- It is hard to compare this method with previous methods: the related work could be more explicit in this regard.
- The discussion over the tightness of the estimated quantities is limited, in more difficult scenarios one could be afraid that bounds will be to loose to be helpful.
- The experiment are limited to 2 dimensional data for most of it and is not detailed enough to provide a good evaluation of the method

**Questions:**

- Could you comment on assumption 3 similarly to what you have done for the other assumptions? It seems that this condition is particularly hard to achieve.
- When using the Sinkhorn divergence, the regularisation terms make interpretation of Wasserstein distance hard is it an issue in practice? Would it be still possible to compute the trace estimate when the source dataset is large?
- Could you detail more the experiment over DonainNet. As your method provide four different terms, would it be possible to modify each of this term to double check how precisely they are captured by your estimator?

---

> ### Author Response · Authors · 2025-11-21
> **Response to Reviewer ns7s (1/2): Contribution & Bound Utility**
>
> We thank you for your insightful review and for recognizing that our paper "addresses an important question" with a clear pipeline. Your technical questions are excellent and allow us to clarify TRACE's distinct novelty and utility.
>
> **1. Clarifying Our Contribution vs. Related Work (Weakness 1).**
>
> We agree this distinction is critical. Our framework differs fundamentally from prior work in its **goal**. We will revise Section 2 to make this explicit.
> *   **Classic DA Theory** (e.g., Ben-David et al., 2010; Redko et al., 2017) aims to bound the **target-domain risk of a single model** (e.g., $R_{\tilde{P}}(Q)$).
> *   **TRACE (Ours)** addresses the MLOps problem of safe model replacement by analyzing the **source-domain risk change between two models**, $|R_P(Q) - R_P(\tilde{Q})|$.
>
> This is a different and previously unaddressed problem. While we use tools like OT/MMD, we repurpose them for attribution within a novel two-model, source-centric decomposition. When replacing a production model, practitioners need to understand *why* performance changed on their core user base, not just predict performance on a new domain. TRACE provides this missing diagnostic lens.
>
> | **Framework** | **Problem Setting** | **Quantity Analyzed** | **Key Terms in Bound** |
> | :--- | :--- | :--- | :--- |
> | Ben-David et al. (2010) | Predict single model's target-domain risk | Target Risk ($R_{\tilde{P}}$) | Domain distance ($\mathcal{H}\Delta\mathcal{H}$), source risk, joint error |
> | Redko et al. (2017) | Predict single model's target-domain risk | Target Risk ($R_{\tilde{P}}$) | Domain distance ($W_1$), source risk |
> | **TRACE (Ours)** | **Diagnose risk change from model update** | **Source Risk Change ($\mid \Delta R_P \mid$)**| **Generalization Gaps, Model Disagreement, Data Shift** |
>
> **2. On Bound Tightness and Practical Utility (Weakness 2).**
>
> This is a crucial point, and we agree that the absolute numerical value of the bound can be loose. Our core thesis is that the structure of the bound is more important than its tightness. TRACE's primary value is in attribution and ranking, where its components are highly useful if they correlate strongly with the true risk change. Our work validates this:
> *   **Theoretical Evidence (Asymptotic Sharpness):** Our Ridge Regression analysis (Sec. 4) proves that both the **true risk difference** and the dominant term in the **TRACE bound** scale as $O(\Vert\mu\Vert)$ with shift magnitude. This shows the bound is **asymptotically sharp** in its scaling, capturing the correct trend.
> *   **Empirical Evidence (Ranking Power):** Our experiments show the TRACE score has a very strong rank correlation with $|\Delta R|$ (Spearman's $\rho \approx \mathbf{0.94}$ in the deployment gate). This strong ranking power enables our gate to achieve near-perfect AUROC, proving its practical utility. We will add a discussion emphasizing TRACE as a "calibrated diagnostic" rather than a "tight certifier."
>
> ***We address the reviewer's specific questions about experiments and technical assumptions in our follow-up comment (2/2) due to character limits.***

---

> ### Author Response · Authors · 2025-11-21
> **Response to Reviewer ns7s (2/2): Experimental & Technical Details**
>
> Continuing our response to Reviewer ns7s:
>
> **3. On Experimental Scope and Detail (Weakness 3 & Question 3).**
>
> We apologize for the lack of clarity and respectfully clarify a misunderstanding. Our evaluation includes **both** low-dimensional synthetic data (for controlled validation) **and** high-dimensional, large-scale vision benchmarks.
> *   **Sec 6.2 & 6.3 (DomainNet):** These experiments use **high-dimensional (2048-d) ResNet-50 features** from a standard, large-scale vision benchmark (~50k images per domain). This is not 2D data, as detailed in Appendices F.7 and F.8.
>
> We will revise Section 6 to better highlight the scale of our benchmarks. Regarding your excellent suggestion to "modify each term," you are correct that this is a crucial validation step. We have revised the paper to make it clear that our synthetic experiments were designed to do exactly this. We added a new paragraph in **Appendix F.6** ("Validating Individual TRACE Components") that details this controlled experimental design. We have also added a pointer to this new section in the main text (Sec. 6.2) for clarity.
>
> Our methodology, now detailed in the appendix, is as follows:
> *   **Isolating the Data Shift (COSP) Term:** To validate this term, we fix the model training procedure and apply geometric data shifts of increasing magnitude. This directly modulates the data shift term, and we show that it correctly tracks the corresponding increase in risk, validating its role.
> *   **Isolating the Model Change Term:** To validate this term, we fix the data distribution and modulate model instability by varying training hyperparameters (e.g., learning rate). This keeps the data shift term constant while causing the *Model Change* term to vary. We verify that its increase correlates with a rise in $|\Delta R|$, confirming its power to diagnose model instability.
>
> These controlled experiments confirm that the components of TRACE successfully isolate and quantify their corresponding sources of risk. Thank you for pushing us to make this crucial aspect of our evaluation more explicit.
>
> **4. On Assumptions and Scalability (Questions 1 & 2).**
> *   **On Assumption 3 (Lipschitz Loss):** This is a standard and mild assumption. For the widely used softmax cross-entropy loss, the function is globally Lipschitz with respect to its logit inputs, with a constant $L_\ell \le \sqrt{2}$. This holds for typical deep learning models. We will add this justification to Section 3.
> *   **On Sinkhorn and Scalability:** We use the **debiased Sinkhorn divergence** for a stable approximation of $W_1$. TRACE scales well to large datasets via standard techniques: (1) **Subsampling** for OT/MMD computation and (2) **GPU-acceleration** via libraries like GeomLoss. We will add these details to the appendix.

---

> > ### Comment · Reviewer_ns7s · 2025-11-24
> >
> > Dear Authors,
> >
> > Thank you for the detailed rebuttal. Would you mind explain further what you find in Appendix F.6? How should I interpret Table 3? I am a bit surprised by the clarification on the related work with only a 2010 paper and a 2017 one, which is quite sparse for ML standards. I am not able to judge this and I would be happy to have other reviewers opinion on this as well. Finally, will you update the paper regarding Sinkhorn computation details?
> >
> > Thank you.

---

> > > ### Author Response · Authors · 2025-11-25
> > > **Response to Reviewer ns7s**
> > >
> > > Dear Reviewer ns7s,
> > >
> > > Thank you for your continued engagement. We are happy to clarify the interpretation of the validation experiments in Appendix F.6 and the scope of our related work.
> > >
> > > **1. Correction: Appendix Referencing**
> > >
> > > We sincerely apologize for a referencing error in our initial response. When you first read our rebuttal, we had mistakenly written "Appendix A.6". We have since corrected this in our posted response. To clarify the correct references:
> > > - The high-dimensional (2048-d) DomainNet experiments are detailed in **Appendices F.7 and F.8**.
> > > - The validation experiments discussed below are in **Appendix F.6**.
> > >
> > > Thank you for your patience with this correction.
> > >
> > > **2. Interpretation of Appendix F.6 and Table 3**
> > >
> > > Appendix F.6 details the experimental protocol for our **synthetic "controlled worlds"** (Blobs-Shift and Moons-Warp), where we sweep shift severity to validate that TRACE components behave as designed.
> > >
> > > **How to interpret Table 3:**
> > > You asked how to read Table 3. Here's a concrete walkthrough using the Moons-Warp experiment:
> > >
> > > - **Mild Shift ($\alpha=0.25$):** $|\Delta R| \approx 0.01$, $\widehat{\mathcal{B}} \approx 0.81$. Crucially, comparing to the baseline ($\alpha=0$), the **Model Change** term remains stable ($\approx 0.48$ vs. 0.50 at baseline), while the **Empirical Shift** term rises (from 0.06 to 0.32) to reflect the geometric change.
> > > - **Severe Shift ($\alpha=2.0$):** $|\Delta R| \approx 0.33$ (~30x increase), $\widehat{\mathcal{B}} \approx 21.5$. Both **Empirical Shift** and **Model Change** contribute, signaling an unsafe update.
> > >
> > > **Key pattern:** $\widehat{\mathcal{B}}$ monotonically tracks $|\Delta R|$ across shift regimes, and attribution terms isolate their intended factors. The diagnostic reliably distinguishes safe from dangerous updates.
> > >
> > > **3. Clarification on Related Work**
> > >
> > > We appreciate the opportunity to clarify. In our rebuttal, we cited Ben-David (2010) and Redko (2017) specifically to highlight the **theoretical distinction** in problem formulations (Target Risk Bound vs. Source Risk Change), as these remain the foundational works establishing the frameworks we build upon.
> > >
> > > Our paper cites a broad range of ML literature (Section 2), including:
> > > - **Discrepancy Measures:** Mansour et al. (2009), Cortes et al. (2010), and Zhang et al. (2019).
> > > - **Optimal Transport:** Courty et al. (2017) and Shen et al. (2018) on Wasserstein DA.
> > > - **Shift Diagnosis:** Recent empirical works like Garg et al. (2021) and Baek et al. (2022).
> > >
> > > We position our specific theoretical contribution against these seminal works to illustrate why a new decomposition is needed for the model replacement problem.
> > >
> > > **4. Sinkhorn Details**
> > >
> > > Yes, we confirm that we have updated the manuscript (**Appendix F.2**) to include the precise details of the Sinkhorn computation. We specify the use of the debiased Sinkhorn divergence to correct for entropic bias, and the use of log-domain stabilization for numerical stability on GPUs.
> > >
> > > We hope this clarifies the interpretation of Table 3 and the context of our work. Please let us know if you would like us to elaborate on any aspect further.
> > >
> > > Best regards,\
> > > The Authors

---

> > > > ### Comment · Reviewer_ns7s · 2025-11-27
> > > >
> > > > Thank you for your detailed answers, which helped me return to the paper. I also read the other reviews and comments, and I believe you addressed my concerns adequately. I still believe that the presentation of this paper could be significantly improved to help readers get the most from it.
> > > >
> > > > My current understanding of the paper is that even though the mathematics behind it are quite simple, and the assumptions may sometimes be a bit too strong, it can still be an interesting method when applicable. The fact that most reviewers have low confidence might also come from the novelty of the method.
> > > >
> > > > Hence, I decided to conclude this discussion by raising my score to weak accept while keeping my low confidence. Maybe this is not a favor to the authors, as the paper would clearly have better future if the presentation were improved, which is more likely to happen if the paper is rejected.

---

> > > > > ### Author Response · Authors · 2025-11-28
> > > > > **Response to Reviewer ns7s**
> > > > >
> > > > > Dear Reviewer ns7s,
> > > > >
> > > > > We sincerely thank you for raising your score and for your time throughout this discussion. We appreciate your constructive feedback on the presentation and the applicability of the method. We will incorporate your suggestions to ensure the final manuscript is as clear and accessible as possible for the community.
> > > > >
> > > > > Best regards,\
> > > > > The Authors

---

> ### Comment · Reviewer_Tg6A · 2025-11-28
>
> I think the concern from `reviewer ns7s` is very reasonable. Although I am sold by this paper and personally like this kind of logical flow, a more careful presentation can be beneficial to persuade a broader class of readers to leverage the proposed theoretical framework and method confidently.
>
> As a reviewer, I appreciate not only the authors' professionalism, but also that of Reviewers ns7s.

---

> > ### Author Response · Authors · 2025-11-28
> > **Response to Reviewer Tg6A**
> >
> > Dear Reviewer Tg6A,
> >
> > Thank you for your continued support and for following the discussion. We appreciate your kind words regarding the professionalism of the process. We agree with the consensus view that improving accessibility is crucial for broader adoption, and we will take the advice from both you and Reviewer ns7s to heart to refine the presentation in the final revision.
> >
> > Best regards,\
> > The Authors

---

### Official Review · Reviewer_c25K · 2025-11-01

**Soundness:** 3
**Presentation:** 3
**Contribution:** 3
**Rating:** 4
**Confidence:** 2

**Summary:**

In this paper, the authors studied the problem of explaining and bounding risk change when replacing a source-trained model with a new model trained on shifted data under covariate shift. They proposed TRACE (Theoretical Risk Attribution under Covariate-shift Effects), a framework that decomposes the absolute risk change $|\Delta R|$ into four interpretable terms: two generalization gaps, a model-change penalty, and a covariate-shift penalty. The method instantiates each term with computable quantities, using Optimal Transport or MMD to estimate data shift, model sensitivity via input gradients, and the discrepancy between model outputs. They demonstrate TRACE theoretically, showing asymptotic sharpness of the bound in linear settings, and empirically validate it across synthetic and vision benchmarks. They show that TRACE's diagnostic score correlates strongly with true performance degradation.

**Strengths:**

- The idea of providing practical explanations and bounds for risk change under model replacing is interesting and novel.
- The proposed framework is effective and practical, both in theory and in experiments.

**Weaknesses:**

- The introduction is not well motivated. The authors should provide more explanation about why it is important to explain and bound risk change under covariate shift.
- The setting is limited. The authors only consider covariate shift, which enforces a strong assumption on the two distributions.

**Questions:**

- Questions
  - In Case Study 1, does the result ($|\Delta R|$) imply that w1min is a better objective than mmdmin? Are there any other metrics to empirically validate this conclusion?
  - In Case Study 2, what is the definition of harmness?
- Suggestions
  - Figure 1 is too wide. I suggest the authors reduce the width of the figure to fit the column width.

---

> ### Author Response · Authors · 2025-11-21
> **Response to Reviewer c25K**
>
> We thank the reviewer for finding our problem "interesting and novel" and our framework "effective and practical." We appreciate the constructive feedback on the motivation and presentation. We address your concerns and questions below.
>
> **1. On Motivation and the Scope of Covariate Shift (Weaknesses 1 & 2).**
>
> We agree the motivation can be made more explicit. We will add a concrete scenario to the introduction: consider a medical imaging model trained at Hospital A ($Q$). It is replaced by an update ($\tilde{Q}$) trained on data from Hospital B's new scanners. For safety, $\tilde{Q}$ must not degrade performance on Hospital A's original patient population. While standard validation detects *that* performance changed, it does not explain *why*. TRACE answers precisely this: is it the scanner shift, or did the model become unstable during retraining?
>
> Our focus on covariate shift enables us to:
> 1.  **Address Prevalent Failures:** It covers critical scenarios like the sensor upgrade described above.
> 2.  **Ensure Tractability:** It allows us to derive a fully *computable* bound via Kantorovich-Rubinstein duality, enabling the separation of geometric shift from algorithmic instability.
> 3.  **Reveal Hidden Trade-offs:** As Table 1 shows, simple heuristics fail here. Minimizing MMD distance paradoxically harmed performance by increasing instability. Without TRACE's decomposition, the root cause of this failure would be unclear.
>
> While this assumption covers ubiquitous real-world cases, we will acknowledge this as a limitation in our conclusion and discuss the roadmap for extending the framework to label and concept drift.
>
> **2. On Case Study 1: `w1min` vs. `mmdmin` (Question 1).**
>
> This is an excellent question. The purpose of this case study is not to declare `w1min` universally superior, but to demonstrate how **TRACE provides a holistic diagnostic that single metrics miss**. The key insight from Table 1 is the trade-off that only TRACE reveals:
> *   The `mmdmin` policy succeeded by its own metric (MMD distance decreased). An engineer monitoring only data alignment would see this as a success.
> *   However, TRACE's *Model Change* term simultaneously spiked (from 27.7 to 66.5), signaling that the optimization caused severe algorithmic instability.
>
> The ground truth validation is the true risk change $|\Delta R|$ (reported in the table), which confirmed that `mmdmin` indeed performed worse. TRACE correctly diagnosed *why* this happened (instability overriding alignment), whereas `w1min` found a better balance. We will clarify this point in Section 6.
>
> **3. On the Definition of “Harmfulness” (Question 2).**
>
> Thank you for catching this omission. In our deployment gate experiment (Sec 6.4), an update is defined as **"harmful" if the true risk on the source test set increases by more than a predefined threshold $\tau$**. Formally: an update is harmful if $R_P(\tilde{Q}) - R_P(Q) > \tau$. We then treat this as a binary classification task to compute AUROC/AUPRC. We will add this precise definition to the experimental setup in Section 6.4.
>
> **4. On Figure 1's Width (Suggestion).**
>
> We thank the reviewer for pointing this out. We will resize Figure 1 to fit the column width in the revised manuscript.

---

### Official Review · Reviewer_iCBh · 2025-11-03

**Soundness:** 3
**Presentation:** 2
**Contribution:** 2
**Rating:** 6
**Confidence:** 2

**Summary:**

This paper proposes the TRACE, which decomposes the upper-bound of the source risk of models updated on shifted data, enabling the analysis or explanation of the risks with factors. To this end, the authors provide the computable proxies that approximate the theoretical factors, giving practical actions that can give valid diagnosis for models showing performance degradations.

**Strengths:**

- TRACE provides the practical lens that ones can diagnose which factors the model performance degradations are attributed to. Such attributions can be fully computable for each factor, which leads to diagnosis in domain adaptations such as failures by model instability or harmful parameter updates.
- Such diagnosis is theoretically sound and its performances seem effective when tested in real-world scenarios such as adaptations to DomainNet dataset.

**Weaknesses:**

- The overall scope of this paper is limited to the covariate shifts, and it cannot be easily extended to other or more complicated distribution shifts. I'm still concerned how practical it is, even considering that the method is theoretically sound.

**Questions:**

Please see weaknesses.

---

> ### Author Response · Authors · 2025-11-21
> **Response to Reviewer iCBh**
>
> We thank the reviewer for their thoughtful assessment and for recognizing that TRACE provides a "practical lens" that is "effective" in real-world scenarios. We appreciate the concern regarding the restriction to covariate shift. We believe this specific focus is what enables us to derive a diagnostic that is both rigorously computable and highly effective for a ubiquitous class of deployment failures.
>
> 1.  **On the Scope and Importance of Covariate Shift.**
>
>     We appreciate the reviewer's point on the scope. Our analysis is indeed focused on covariate shift, which we chose as a critical first step toward a broader diagnostic framework. This specific focus enabled us to develop a complete, end-to-end solution with a computable analysis for what is arguably one of the most common failure modes in deployed ML systems. Covariate shift ($P(x) \neq \tilde{P}(\tilde{x})$ but $P(y \mid x) \approx \tilde{P}(y \mid \tilde{x})$) occurs whenever input statistics change but the underlying task logic remains constant. This setting is highly practical and covers a wide range of high-stakes transitions:
>
>     *   **Domain Generalization:** E.g., A medical imaging model deployed to a new hospital with a different scanner protocol. The visual statistics change, but the medical diagnosis does not. Our DomainNet experiments demonstrate TRACE's effectiveness in this setting.
>     *   **Environmental Drift:** E.g., An autonomous vehicle encountering different weather conditions. The definition of a "pedestrian" remains the same, but the input distribution shifts significantly.
>     *   **Hardware/Sensor Changes:** Upgrades to sensors or data pipelines that alter input statistics without changing the labels.
>
>     By focusing on this well-defined and critical problem, we provide an end-to-end, actionable solution. We agree that extending TRACE is an exciting future direction and will add a paragraph on **"Limitations and Future Work"** to the Conclusion discussing the roadmap for handling label and concept shifts.
>
> 2.  **On the Practicality and Actionability of TRACE.**
>
>     We appreciate the concern about practicality. TRACE’s value is not in suggesting generic actions, but in providing a holistic diagnostic that prevents the dangerous pitfalls of monitoring single metrics in isolation. Engineers often see a problem like data drift and focus only on fixing that, which can lead to unintended consequences.
>
>     A prime example is our Case Study 1 (Table 1), which simulates this exact scenario:
>
>     *   A team, seeing data drift, implements a policy (mmdmin) to minimize it. By that single metric, the policy is a success, and a standard dashboard would show a green light.
>     *   However, overall performance catastrophically degrades. Without a holistic view, the team would be left guessing why.
>     *   TRACE diagnoses the problem instantly: the mmdmin policy, while fixing one issue, caused a massive spike in the Model Change term (from 27.7 to 66.5), revealing that the root cause was severe model instability from the update process itself.
>
>     The lesson is that focusing on one metric can be dangerously misleading. This ability to de-risk model updates by providing a multi-faceted view is not an isolated anecdote. Our Deployment Gate experiment (Sec. 6.4) validates this at scale: the TRACE score achieves a strong rank correlation ($\rho \approx 0.94$) with the true risk change, translating into a highly reliable automated gate (AUROC $\approx$ 1.0).

---

### Author Response · Authors · 2025-11-21
**Revised Manuscript Uploaded**

We sincerely thank all reviewers for their time and valuable feedback. We are encouraged that the reviewers found that our work addresses an **"important question"** (Reviewer ns7s), is **"interesting and novel"** (Reviewer c25K), provides a **"practical lens"** (Reviewer iCBh), and is presented with **"Exceptional Clarity"** (Reviewer Tg6A). The feedback provides a clear path to strengthen the paper. A central theme is the practicality and scope of TRACE. We will revise the paper to emphasize that TRACE's primary value is as a **practical, ranking-based diagnostic** for the critical MLOps problem of safe model replacement, rather than a tight certifier. Below, we address each reviewer's points in detail.

**We have uploaded a revised manuscript incorporating your suggestions, with major changes highlighted in $\textcolor{blue}{blue}$.**

---

### Comment · Area_Chair_uoSE · 2025-11-27

Dear reviewers,

The authors have provided detailed responses to your reviews. If you have not done so already, I would appreciate if you could let both me and the authors know how these responses impact your assessment of the paper.

Best,

AC

---

### Meta-Review · Area_Chair_PRWv · 2026-01-06

**Summary:**

While TRACE is a competently executed engineering contribution with a plausible practical use case, it does not meet the bar for a top-tier conference due to its limited theoretical novelty, restrictive assumptions, and narrow empirical validation. The work would be better suited for a workshop or a more application-focused venue. For eventual publication, the authors should: (1) provide a more rigorous theoretical analysis, possibly deriving tighter, data-dependent bounds; (2) relax or justify the strong smoothness assumptions; (3) extend the framework and experiments beyond covariate shift; and (4) demonstrate scalability and robustness on larger, more diverse benchmarks.

**Reviewer Scores:**

No

---

### Decision · Program_Chairs · 2026-01-26

Reject